# Concentrations of dissolved dimethyl sulfide (DMS), methanethiol and other trace gases in context of microbial communities from the temperate Atlantic to the Arctic Ocean

**Valérie Gros[1], Bernard Bonsang[1], Roland Sarda-Estève[1], Anna Nikolopoulos[2], Katja Metfies[3], Matthias Wietz[3,4], and Ilka Peeken[3]**

[1]Laboratoire des Sciences du Climat et de l'Environnement, CNRS-CEA-UVSQ, IPSL, 91 191 Gif sur Yvette, France
[2]Norwegian Polar Institute, Fram Centre, 9296 Tromsø, Norway
[3]Alfred Wegener Institute Helmholtz Centre for Polar and Marine Research, 27570 Bremerhaven, Germany
[4]Max Planck Institute for Marine Microbiology, 28359 Bremen, Germany

**Correspondence:** Valérie Gros (valerie.gros@lsce.ipsl.fr)

**Abstract.** CE1 Dimethyl sulfide (DMS) plays an important role in the atmosphere by influencing the formation of aerosols and cloud condensation nuclei. In contrast, the role of methanethiol (MeSH) for the budget and flux of reduced sulfur remains poorly understood. In the present study, we quantified DMS and MeSH together with the trace gases carbon monoxide (CO), isoprene, acetone, acetaldehyde and acetonitrile in North Atlantic and Arctic Ocean surface waters, covering a transect from 57.2 to 80.9° N in high spatial resolution in May–June 2015. Whereas isoprene, acetone, acetaldehyde and acetonitrile concentrations decreased northwards, CO, DMS and MeSH retained substantial concentrations at high latitudes, indicating specific sources in polar waters. DMS was the only compound with a higher average concentration in polar ($31.2 \pm 9.3$ nM) than in Atlantic waters ($13.5 \pm 2$ nM), presumably due to DMS originating from sea ice. At eight sea-ice stations north of 80° N, in the diatom-dominated marginal ice zone, DMS and chlorophyll $a$ markedly correlated ($R^2 = 0.93$) between 0–50 m depth. In contrast to previous studies, MeSH and DMS did not co-vary, indicating decoupled processes of production and conversion. The contribution of MeSH to the sulfur budget (represented by DMS + MeSH) was on average 20 % (and up to 50 %) higher than previously observed in the Atlantic and Pacific oceans, suggesting MeSH as an important source of sulfur possibly emitted to the atmosphere. The potential importance of MeSH was underlined by several correlations with bacterial taxa, including typical phytoplankton associates from the *Rhodobacteraceae* and *Flavobacteriaceae* families. Furthermore, the correlation of isoprene and chlorophyll $a$ with *Alcanivorax* indicated a specific relationship with isoprene-producing phytoplankton. Overall, the demonstrated latitudinal and vertical patterns contribute to understanding how concentrations of central marine trace gases are linked with chemical and biological dynamics across oceanic waters.

## 1 Introduction

Volatile organic compounds (VOCs) and carbon monoxide (CO) are important in atmospheric chemistry as precursors of ozone and secondary organic aerosols, which affect air quality and climate. Despite being a relatively small source compared to anthropogenic emissions (Duncan et al., 2007; Kansal, 2009) and terrestrial vegetation (Guenther et al., 1995), the global oceans are increasingly considered as sources and sinks of CO and VOCs with potential influence on atmospheric chemistry. Biological activities substantially contribute to the dynamics of short-lived VOCs like dimethyl sulfide (DMS) and isoprene. For instance, dimethylsulfoniopropionate (DMSP) produced by phytoplankton and other marine organisms (such as macroalgae CE2, corals and sponges; Jackson and Gabric, 2022, and

references therein) can be metabolized by bacteria into DMS. DMS is rapidly oxidized once emitted to the atmosphere (1 d TS1 mean lifetime, Kloster et al., 2006), then representing a major precursor of sulfated aerosols with radiative impacts by scattering sunlight and constituting condensation nuclei (CCN), potentially cooling the climate through changing cloud microphysics. The role of DMS emissions on climate was first hypothesized by Shaw (1983) and Charlson et al. (1987) and is known as the CLAW hypothesis, which has been discussed and debated since then. There is extensive literature on CE3 the link between DMSP and DMS, the multi-steps of the DMS oxidation to sulfate, and the corresponding impact on CCN. As these processes are beyond the scope of this paper, we refer the reader to the recent review of Jackson and Gabric (2022) and references therein for further information. Alternatively, DMSP can be microbially demethylated into methanethiol ($CH_3SH$, here referred to as MeSH; Kiene, 1996; Kiene and Linn, 2000), whose role in the atmosphere and oceans is poorly characterized to date (Lawson et al., 2020). The oxidation of MeSH by hydroxyl radicals (Tyndall and Ravishankara, 1991; Butkovskaya and Setser, 1999) is estimated to effectively produce $SO_2$ with up to 48 % based on model calculations (Novak et al., 2022). Thus, MeSH is probably an underestimated factor in the marine sulfur cycle.

Isoprene, another important trace gas, can be produced by photosynthesizing organisms over short timescales (a few hours), with potential influence on regional atmospheric chemistry and aerosol formation above biologically active pelagic waters (Bikkina et al., 2014). Photosynthetic cyanobacteria are stronger emitters of isoprene than diatoms, with taxon-specific variability in production (Bonsang et al., 2010; Shaw et al., 2010). Besides direct emission by primary producers, oceanic trace gases can originate from photochemical processes. For instance, isoprene can be photochemically produced at the sea-surface microlayer (Ciuraru et al., 2015). In addition, photodegradation of dissolved organic matter is the main source of CO (Wilson et al., 1970), although laboratory experiments showed a minor contribution of biological activity (Gros et al., 2009).

The oceanic contribution to the budget of oxygenated VOCs (OVOCs) like acetone, acetaldehyde and methanol is also important to consider, since OVOCs may affect the oxidative capacity of the remote atmosphere through tropospheric radicals (Singh, 2004). A recent study has confirmed the importance of air–sea exchange for acetaldehyde, pointing out the lack of oceanic measurements (Wang et al., 2019). Marine waters can be either a local source or sink of OVOCs depending on the region. Acetone and acetaldehyde are considered to originate from photodegradation of dissolved organic carbon (Zhou and Mopper, 1997; Zhu and Kieber, 2019). A positive net flux of acetone is generally observed in biologically productive areas such as tropical upwelling zones, whereas high-latitude and oligotrophic waters represent sinks (Lawson et al., 2020). As OVOCs mainly origi-

nate from terrestrial sources, their air–sea fluxes can also be a net deposition, when their marine atmospheric concentrations are directly influenced by air masses originating from continents (Phillips et al., 2021). Furthermore, OVOCs can show seasonal variation (Davie-Martin et al., 2020).

Linking the dynamics of (O)VOCs and trace gases to primary production and microbial distribution helps to understand fundamental couplings between biological, oceanic and atmospheric processes. This is particularly important in the Arctic Ocean, which warms 2 to 3 times CE4 faster than the global average (Schmale et al., 2021 and references therein). Ocean warming concurs with physical, biological and photochemical variability, subsequently affecting the coupling between ocean and atmosphere. Importantly, sea-ice melt influences VOCs production, e.g. through increased primary production in ice-free waters (Arrigo and van Dijken, 2015), release of ice algae and their substrates (Fernández-Méndez et al., 2014), as well as higher gas exchange at the ocean-atmosphere interface (Lannuzel et al., 2020) when ice-free areas expand. This influence of sea ice has been shown for DMSP, DMS, isoprene, acetone and acetaldehyde in the Canadian Arctic (Galindo et al., 2014; Wohl et al., 2022; Galí et al., 2021). Concurrent changes in phytoplankton distribution can amplify these dynamics, for instance via the northward expansion of the coccolithophorid *Emiliania huxleyi* by Atlantic currents (Hegseth and Sundfjord, 2008; Oziel et al., 2020) and changing bloom phenologies (Nöthig et al., 2015; von Appen et al., 2021). As phytoplankton and bacterial distribution are often linked, these dynamics subsequently affect the heterotrophic food web. The bacterial families *Rhodobacteraceae* and *Flavobacteriaceae* are frequently abundant during phytoplankton blooms, contributing to the degradation of algal organic matter and the conversion of DMSP into DMS and MeSH (Moran et al., 2012; Moran and Durham, 2019). Campen et al. (2022) have recently emphasized the need to better link bacterial distribution with DMS and CO metabolism. In addition, the production and degradation of isoprene could be an important, yet understudied contribution to biogeochemical cycles (Carrión et al., 2020; Rodríguez-Ros et al., 2020; Simó et al., 2022).

As Atlantic characteristics are observed to expand northward with climate change (Polyakov et al., 2020), it is important to contextualize marine VOCs, trace gases and microbes in Arctic versus temperate Atlantic waters. Here, we report concentrations of DMS, MeSH, isoprene, CO, acetone, acetaldehyde and acetonitrile in context of microbial distribution across the North Atlantic and Arctic oceans. During the Transitions in the Arctic Seasonal Sea Ice Zone (TRANS-SIZ) campaign onboard RV *Polarstern* from early spring to summer 2015, we continuously measured these compounds in surface waters between 57 to 80° N, and additionally over vertical profiles from the surface to 50 m depth in the ice-covered region north of Svalbard (Fig. 1). The main objective was to document the concentrations and spatial variability of trace gases, specifically the ratio between MeSH and

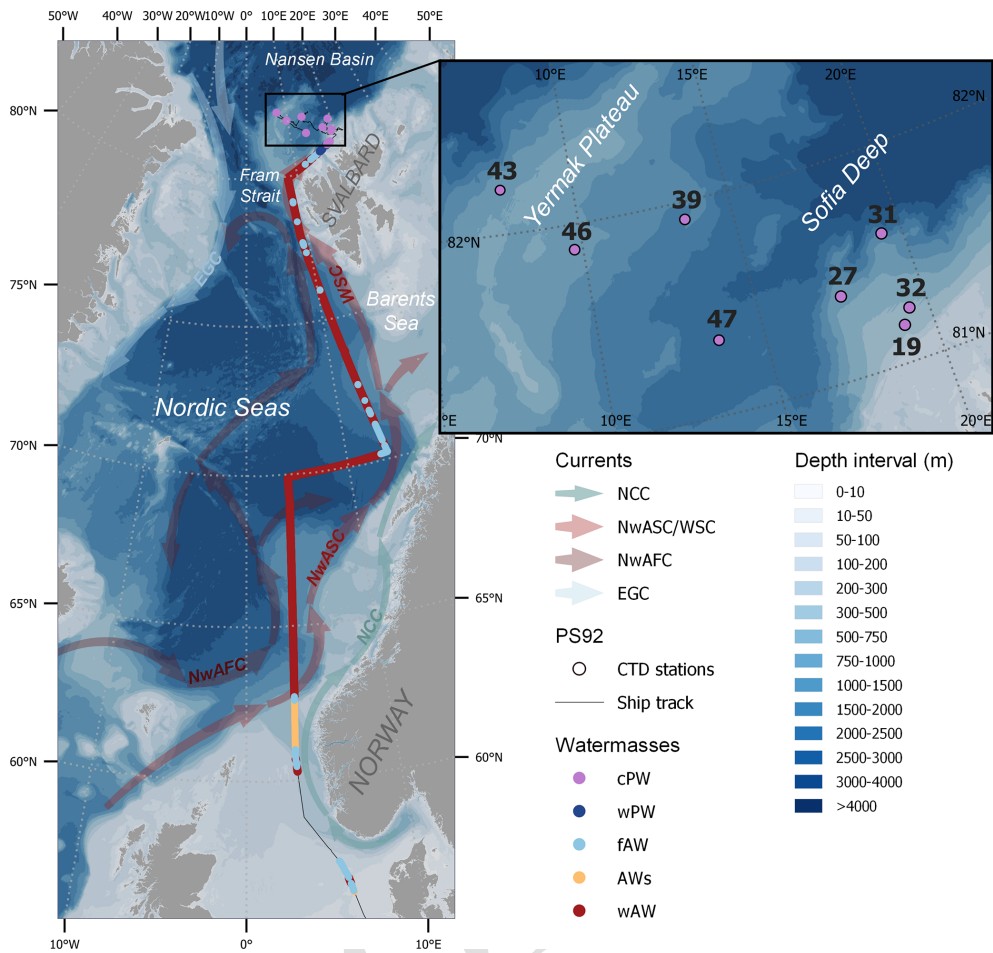

**Figure 1.** Ship track coloured by water mass: "regular" warm Atlantic Water (wAW), coastal-influenced Atlantic Water CE5 with low salinity (AWs), freshened and cooled Atlantic Water (fAW), warm Polar Water (wPW) and cold Polar Water (cPW), determined according to the temperature and salinity criteria in Table 1. Surface measurements were done continuously between 57 and 81° N, while vertical profiles were sampled at eight sea-ice stations (black insert and Table S1). The background map shows bathymetry (GEBCO, 2022) and a schematic overview of the major currents influencing the surface waters in the study area, as adapted from Skagseth et al. (2022): the Norwegian Atlantic Slope Current (NwASC), West Spitsbergen Current (WSC), Norwegian Atlantic Front Current (NwAFC), Norwegian Coastal Current (NCC) and East Greenland Current (EGC).

DMS, in context of phytoplankton biomass, bacterial diversity and water masses. To our knowledge, this is the first survey of MeSH in the Arctic Ocean, shedding light on its biogeochemical role in an area of rapid climate change.

## 2 Material and methods

### 2.1 Campaign description and oceanographic parameters

Water samples were collected during the TRANSSIZ (PS 92 – ARK XXIX/1) cruise onboard RV *Polarstern* between 19 May and 28 June 2015. The cruise started in Bremerhaven, Germany, and ended in Longyearbyen, Svalbard (Fig. 1), as described in detail by Peeken (2016).

Along the ship track between 19 and 27 May, temperature, salinity and chlorophyll *a* (Chl *a*) fluorescence in the surface water layer (6 m depth) were continuously recorded with the FerryBox System of the Helmholtz-Zentrum Hereon, and extracted from the FerryBox database every 2 min. The instrument performs a self-cleaning routine every day, including acid washing and freshwater rinsing. In addition, sensor behaviour is controlled by staff members of *Polarstern*, although drift of the sensors is rarely observed (for details see Petersen, 2014). The sensor for Chl *a* was a submersible fluorometer (Turner Designs, Sunnyvale, CA, USA) with excitation/emission wavelengths of 460 and 620–715 nm respectively.

After 27 May, eight ice stations (number 19, 27, 31, 32, 39, 43, 46 and 47; for details see Table S1 in the Supplement) were carried out over the continental shelf north of

Svalbard and over the Yermak Plateau (Fig. 1; Table S1). At each ice station, the ship was anchored to an ice floe at drift for approximately 36 h. While carrying out ice work on the port side, winch-operated instruments were deployed in the open water on the starboard side to record biological and biogeochemical variables including trace gases and phytoplankton pigments. Except for the first ice station (70 % ice cover and still some leads present), the stations were conducted in almost 100 % ice cover (Massicotte et al., 2019), with the sampling taking place in small leads. All ice stations were 50 to 250 km away from the ice edge and open water (Dybwad et al., 2021). A detailed study of nutrients, marker pigments and protist microscopy classified the Yermak Plateau stations (39, 43, 46) to be in a pre-bloom phase, while all other stations were in a bloom phase (Dybwad et al., 2021). During the ice stations, discrete seawater samples for trace gas and phytoplankton analyses were collected at six water depths between 0.5 and 50 m depth using 12 L Niskin bottles on the ship-CTD (conductivity, temperature, depth) water-sampling carousel. For trace gases, these samples were transferred to 1 L light-proof glass flasks for direct analysis on board. Values for temperature and salinity were provided by the CTD bottle data file for each station (Nikolopoulos et al., 2016).

Temperature and salinity from both types of measurements were used to classify the sampled water masses based on the criteria applied in Tran et al. (2013; Table1).

## 2.2 Biological measurements

### 2.2.1 Pigment analysis

For pigment analysis with high pressure liquid chromatography (HPLC), seawater samples (1–2 L) were taken from the Niskin bottles on the ship-CTD from six depths in the upper 50 m (Table S1). All samples were processed within few hours after collection.

Sample handling and pigment measurements were carried out as described in Tran et al. (2013). The FerryBox/surface Chl $a$ data were calibrated against surface Chl $a$ concentrations derived from the Niskin bottles ($R^2 = 0.83$, see Fig. S1 in the Supplement TS2). The taxonomic composition of phytoplankton was calculated from marker pigments using the CHEMTAX approach (for details see Wollenburg et al., 2018), distinguishing diatoms, *Phaeocystis*-type haptophytes, prasinophytes, chlorophytes, dinoflagellates, cryptophytes, chrysophytes and coccolithophorid-type haptophytes. The contribution of each group was expressed as Chl $a$ concentration.

### 2.2.2 Bacterial community analysis

A total of 34 seawater samples for bacterial community analysis were collected along the transect (Table S2) at a depth of $\sim 10$ m using the AUTOFIM system (Metfies et al., 2016, 2020), which is installed at the bow of the ship next to the CE6

pump system intake. Per sampling event, 2 L of seawater were filtered onto polycarbonate filters with 45 mm diameter and 0.4 µm pore size (Millipore; USA) at 200 mbar. Filters were stored at $-80\,°C$ until DNA extraction in the home laboratory using the NucleoSpin Plant II kit (Macherey-Nagel, Germany) according to the manufacturer's instructions. Bacterial 16S rRNA gene fragments were amplified using primers 515F–926R (Parada et al., 2016) according to the 16S Metagenomic Sequencing Library Preparation protocol (Illumina, San Diego, CA). Amplicon gene libraries were sequenced using Illumina MiSeq technology in $2 \times 300$ bp paired-end runs at CeBiTec (Bielefeld, Germany). Raw sequence files have been deposited in the European Nucleotide Archive under accession number PRJEB50492, using the data brokerage service of the German Federation for Biological Data (GFBio) in compliance with MIxS standards. The amplicon analysis workflow is described in Sect. S2 in the Supplement, with bioinformatic code and data files documented under https://doi.org/10.5281/zenodo.7609524 (Wietz et al., 2023). Briefly, after primer removal using cutadapt (Martin, 2011), reads were classified into amplicon sequence variants (ASVs) using DADA2 (Callahan et al., 2016) and taxonomically classified using the Silva v138 database (Quast et al., 2012). We obtained on average 85 000 quality-controlled, chimera-filtered reads per sample (Table S2) sufficiently covering community composition (Fig. S2). Nonmetric multidimensional scaling was performed to determine bacterial community variability along the transect. Associations between the abundance of bacterial ASVs and environmental parameters were determined via Holm-corrected Spearman's correlations. Only correlations > |0.4| were considered, and only if higher than correlations with latitude to omit indirect signals due to geographical variability.

## 2.3 Trace gas measurements

Carbon monoxide and VOCs dissolved in seawater were measured in real-time along the transect using samples from the FerryBox water intake (6 m depth). Seawater was delivered by the ship membrane pump to the laboratory for continuous injection into an online water extraction device (OLWED; Sect. S3, Fig. S3). Furthermore, we measured trace gas concentrations from 0.5 to 50 m depth at the eight ice stations, analysing all samples within few hours after collection. Possible artefacts from storage have been investigated in a previous experiment in the same area (Tran et al., 2013), showing no significant losses of low molecular weight VOCs and a slow decrease for CO during the first 4 h (Xie and Zafiriou, 2009; Tolli and Taylor, 2005). As no cross calibration was made between transect and Niskin measurements, possible differences between on-line and off-line measurements could not be evaluated.

### 2.3.1 PTRMS measurements

VOCs were quantified using a high-sensitivity proton-transfer mass spectrometer (PTRMS, Ionicon Analytik) developed by Lindinger and Jordan (1998) and since then widely used (reviewed by Blake et al., 2009). The measurement principle of PTRMS is based on the soft chemical ionization of VOCs by proton transfer, which is possible for all compounds with a proton affinity higher than water, giving access to a variety of VOCs (Blake et al., 2009). During the campaign, air from the headspace was continuously sampled by the PTRMS through a 1/8 in. Teflon-PFA line at a flow rate of about $60\,mL\,min^{-1}$ using standard parameters, i.e. $60\,°C$ (inlet and drift tube temperature), 600V drift tube and 2.2 mbar drift tube pressure (with a corresponding E/N of 132 Townsend). The estimated residence time of about 30 s should prevent degradation or adsorption of extracted gases in the system. Furthermore, a series of standards were measured under the same experimental conditions, showing high linearity in the system's response supporting the absence of artefacts. Measurements were typically performed every 2.5 min, except between 61.1 and 65.3° N, where measurements were performed every 10 min for approximately 24 h to scan a wider range of masses ($m/z$). This step was needed to select the compounds of interest, i.e. those showing a signal above the detection limit. About 25 masses ($m/z$) were selected to be further monitored (with dwell times from 1 to 20 s). Here, we present the results for the compounds with the most significant variability: isoprene ($m/z$ 69), dimethyl sulfide ($m/z$ 63), methanethiol ($m/z$ 49), acetone ($m/z$ 59), acetaldehyde ($m/z$ 45) and acetonitrile ($m/z$ 42). The only small fragmentation from soft ionization allows direct measurements of compounds at their corresponding $m/z + 1$. Although we cannot rule out higher molecule fragmentation, interferences from other compounds are likely negligible for the masses presented in this study (Blake et al., 2009; Yuan et al., 2017). An exception could be isoprene, as it can contain fragments of 2-methyl-3-buten-2-ol (MBO) or cyclohexanes. However, during the period when the PTRMS measured in scan mode, MBO mass ($m/z$ 87) was uncorrelated ($R^2 = 0.02$) with $m/z$ 69. In addition, the good correlation ($R^2 = 0.77$) between isoprene and Chl $a$ across vertical profiles (Fig. S6) confirms that the measured $m/z$ 69 can be mainly attributed to isoprene. For acetone, the signal corresponds to "acetone + propanal", but propanal can be neglected and $m/z$ 59 be considered as acetone (de Gouw and Warneke, 2007). The PTRMS used for this campaign was used the year before on a field campaign, and some of its characteristics are described in Zannoni et al. (2016). The calibration procedures for gas and water phases are given in Sect. S4, Fig. S4a and S4b. The measurement uncertainty is estimated at $\pm 20\%$, taking into account errors related to standard gas, calibrations, blanks, reproducibility and linearity (see Sect. S4). The overall uncertainty for dissolved VOC measurements was estimated at $\pm 30\%$, except for MeSH. Due to the missing direct calibration of MeSH, its concentration was possibly underestimated by up to 1.5 times (see Sect. S4). Therefore, reported concentrations presented here have to be considered as lower limit for MeSH.

### 2.3.2 CO measurements

CO was measured using a custom-made gas chromatograph directly coupled to the extraction cell, equipped with a hot mercuric-oxide detector operating at $265\,°C$ (RGA3, Trace Analytical, Menlo Park, CA). The system comprised two 1 mL nominal volume stainless-steel injection loops for samples and calibration respectively, previously calibrated in the laboratory. The chromatographic procedure used a pre-column (0.77 m length, 0.32 cm outer diameter, containing Unibeads 1S 60/80 mesh) and an analytical column (0.77 m length, 0.32 cm outer diameter, containing molecular sieve 13X 60/80 mesh) both heated to $95\,°C$. Air from the headspace of the extraction cell and standard gas were alternately injected into the chromatograph, each sample being directly calibrated with the previous injection of the standard gas. The standard gas consisted of CO diluted in synthetic air at a nominal concentration of 200 ppbv. The CO retention time was 1.5 min, and a complete chromatogram ran for 2.5 min. The overall accuracy of the measurement was about 5%. More details about CO measurements are described in Gros et al. (1999) and Tran et al. (2013).

## 3 Results

### 3.1 Latitudinal variability in surface waters from 57 to 80° N

Along the transect, surface measurements of temperature, salinity and Chl $a$ were performed across five different water masses: warm Atlantic Water with low salinity (AWs), "regular" warm Atlantic Water (wAW), freshened and cooled Atlantic Water (fAW), cold Polar Water (cPW), and warm Polar Water (wPW) as defined in Table 1. The major part of the transect, from 63 to 80° N, occurred in wAW (Fig. 1). Atlantic Water with lower salinity (AWs and fAW) was encountered in the vicinity of the Norwegian Coastal Current, which carries water masses influenced by river run-off: AWs at 60.6–62.3° N (with fAW in the mixing zones) and fAW around 70–72° N. Fresher mixed products (fAW) were also intermittently encountered west of Svalbard (where AW meets fjord/coastal water masses), as well as in the marginal ice zone where AW mixes with and gradually subducts under fresher polar waters. Polar Water (PW) only occurred north of 80° N in the Nansen Basin. Surface temperature steadily decreased northwards, from 8 to below 0 °C in the ice-covered region $> 80°$ N (Fig. 2). The slight deviation around 70° N corresponds to the shifting cruise track towards Tromsø due to a medical evacuation event (Fig. 1). Chl $a$ con-

**Table 1.** Mean values and standard deviation for concentration of trace gases in five different water masses along the transect (see Fig. 1 for exact areas), and from surface samples at eight sea-ice stations north of 80° N. $S$: salinity; BDL: below detection limit. * In italics: data from (Tran et al., 2013) in the same area but in June–July 2010, i.e. 1 month later in summer. Due to sensor failure of temperature and salinity, the records start at 60° N.

| | Acetonitrile (nM) | Acetaldehyde (nM) | Acetone (nM) | DMSy (nM) | Methanethiol (nM) | MeSH/ (MeSH + DMS) % | Isoprene (pM) | CO (nM) |
|---|---|---|---|---|---|---|---|---|
| Coastal-influenced/low-salinity Atlantic Water (AWs; $\theta > 5\,°C$, $S < 34.4$) | $1.11 \pm 0.55$ | $19.67 \pm 7.96$ | $23.34 \pm 12.77$ | $15{,}65 \pm 6.96$ | $0.84 \pm 0.65$ | $5.6 \pm 7.1$ | $2{,}55 \pm 0.84$ *23.4 ± 3.10** | $10.70 \pm 3.07$ *2.50 ± 1.70** |
| Warm Atlantic Water (wAW; $\theta > 2\,°C$, $S > 34.9$) | $0.53 \pm 0.23$ | $4.84 \pm 4.03$ | $2.36 \pm 5.88$ | $11{,}75 \pm 6.97$ | $2.89 \pm 1.52$ | $21.9 \pm 8.7$ | $1{,}38 \pm 0.70$ *42.5 ± 49.6** | $5.86 \pm 2.77$ *3.3 ± 2.2** |
| Freshened Atlantic Water (fAW; $\theta > 1\,°C$, $34.4 < S < 34.9$) | $0.94 \pm 0.40$ | $9.84 \pm 5.60$ | $14.56 \pm 10.80$ | $13.05 \pm 8.83$ | $3.26 \pm 1.49$ | $20.7 \pm 10.6$ | $2{,}66 \pm 1.51$ *24.8 ± 19.1** | $10.17 \pm 5.89$ *3.4 ± 2.4** |
| Cold Polar Water (cPW; $\theta < 0\,°C$, $S < 34.7$) | $0.32 \pm 0.13$ | $0.98 \pm 2.27$ | BDL | $30.03 \pm 9.26$ | $2.80 \pm 0.76$ | $9.1 \pm 2.3$ | $1{,}22 \pm 0.47$ | $5.00 \pm 2.82$ |
| Warm Polar Water (wPW; $\theta > 0\,°C$, $S < 34.4$) | $0.21 \pm 0.09$ | $0.30 \pm 0.86$ | BDL | $34.65 \pm 8.46$ | $3.49 \pm 0.29$ | $9.6 \pm 2.0$ | $1{,}06 \pm 0.28$ | $7.81 \pm 2.08$ |
| Polar Water (PW) (cPW + wPW) | $0.30 \pm 0.13$ | $0.84 \pm 2.05$ | BDL | $31.19 \pm 9.29$ | $2.96 \pm 0.74$ | $9.2 \pm 2.2$ | $1{,}19 \pm 0.44$ *14.5 ± 11.5** | $5.88 \pm 2.91$ *6.5 ± 3.2** |
| Surface water at sea-ice stations > 80° N (range) | $0.28 \pm 0.12$ (0.15–0.47) | $7.24 \pm 4.43$ (0.27–14.23) | $2.29 \pm 2.79$ (0–6.93) | $11.22 \pm 10.91$ (1.64–31.90) | $0.13 \pm 0.17$ (0.02–0.53) | | $3.23 \pm 2.07$ (0.90–7.25) | $1.45 \pm 1.67$ (0.24–4.26) |

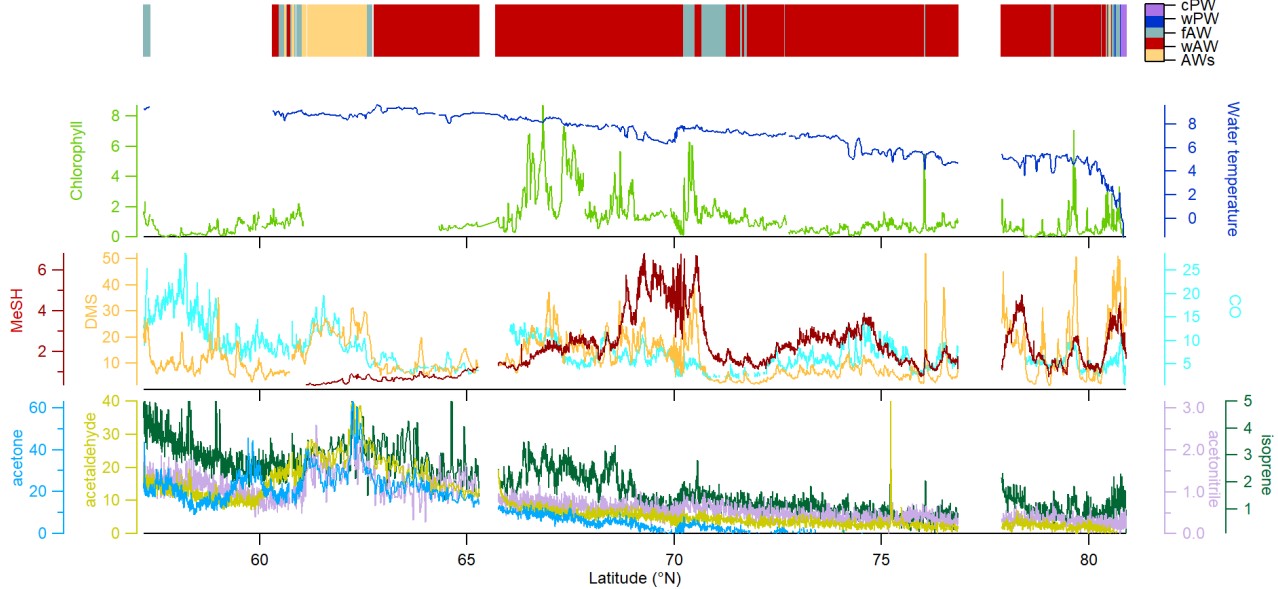

**Figure 2.** Latitudinal variability of acetone (nM), acetaldehyde (nM), acetonitrile (nM), isoprene (pM), DMS (nM), MeSH (nM) and CO (nM) between 57.2 to 80.9° N in relation to Chl $a$ ($\mu g\,L^{-1}$) and water temperature (°C). Due to sensor failure, temperature records are missing until $\sim 61$° N. The colored horizontal bar on top illustrates the encountered water masses (Fig. 1). Values below 3 nM are below the detection limit for acetone and acetaldehyde (see Sect. S4). Data are available at Peeken et al. (2023a).

centrations peaked at the beginning of the transect, with over-all five areas where concentrations exceeded $1\,\mu g\,L^{-1}$ indicating increased phytoplankton abundances: $\sim 60$ to 61° N (up to $2\,\mu g\,L^{-1}$), several locations $> 66$° N ($> 6$–$8\,\mu g\,L^{-1}$), and three peaks ($> 5\,\mu g\,L^{-1}$) at 76, 78 and 79.5° N respectively. Within the marginal ice zone ($> 80$° N) Chl $a$ concentrations reached up to $3\,\mu g\,L^{-1}$.

### 3.1.1 Trace gas distribution

The oxygenated gases acetone and acetaldehyde strongly decreased with higher latitude and lower water temperature, being below or close to the detection limit north of 70° N. Nevertheless, we observed a 2- to 3-fold increase between 61 and 65° N. Acetone varied from 20 to 25 nM between 57

to 65° N, decreasing to 0.1 nM near 80° N. A maximum of 40 nM between 60 and 65° N covaried with higher Chl $a$ concentrations, plus a second minor peak between 77 and 79° N. Similar latitudinal trends occurred for acetaldehyde and acetonitrile. Acetaldehyde decreased from 15 nM in the temperate Atlantic to 0.5–3 nM in the Arctic Ocean, with a peak of approx. 40 nM between 60–65° N. Acetonitrile decreased from 1.5 nM to 0.1–0.5 nM, with a second maximum of approx. 2 nM between 60 and 65° N. Isoprene decreased from 4–5 pM at 57° N to 0.3–1.5 pM at 80° N, with three additional maxima along the transect. Opposed to the other trace gases, isoprene slightly increased again north of 80° N, albeit at a much lower concentration compared to lower latitudes.

CO, DMS and MeSH displayed different patterns, retaining high but variable concentrations at high latitudes. CO concentrations varied between 2 and 30 nM, with several peaks covarying with Chl $a$ at 62.5, 67 and 77.6° N. DMS ranged from ∼2 to 50 nM, with peaks at 61–63, 66–70.5° N and a maximum of 60 nM at 80° N. MeSH varied from 0.1 to 7 nM, with peaks near 70 and between 73–75° N.

### 3.1.2 Bacterial communities in the environmental context

We performed 16S rRNA amplicon sequencing to characterize bacterial community structure in context of latitude, water temperature and trace gas concentrations. Correspondent to the known microbial differences between temperate and polar oceans (Sunagawa et al., 2015), communities substantially varied by latitude and temperature. These factors explained 43 % of bacterial variability (PERMANOVA; $p < 0.001$). Accordingly, communities markedly varied between Atlantic and polar waters north of 80° N (Fig. 3a). Several correlations between trace gases, Chl $a$ and the abundance of specific ASVs (Fig. 3b) suggests bacterial linkages with phytoplankton and VOC dynamics. Correlations were both positive and negative, sometimes differing within single genera. For instance, one SAR92-ASV positively correlated with MeSH, whereas another SAR92-ASV negatively correlated with DMS. For MeSH and its ratio to DMS, correlations differed between *Pseudofulvibacter*, NS10, OM75, *Yoonia-Loktanella* and *Ascidiaceihabitans* ASVs (negative) versus SUP05 (positive). *Synechococcus* and an unclassified cyanobacterial ASV were negatively correlated with MeSH. DMS positively correlated with ASVs from *Aurantivirga* and SAR11 clade Ia. Two ASVs from the NS9 clade were unique in their correlations with acetone and acetonitrile. Furthermore, several ASVs from *Thalassolituus* and *Alcanivorax* (negative), NS5 and *Polaribacter* (positive) correlated with Chl $a$ and isoprene.

### 3.2 Vertical under-ice profiles north of 80° N

In the ice-covered region north of Svalbard, we performed vertical under-ice profiles at eight stations instead of the continuous surface seawater measurements (Figs. 1, 4, and S5, Sect. 2.1). To connect latitudinal and vertical records, we compared the cPW values measured along the transect with the surface values (0.5 m depth) from the vertical profiles (Table 1, Fig. 4). This revealed marked differences in trace gas concentrations compared to polar water masses along the transect, except for acetonitrile. Acetaldehyde concentrations (0.3 to 14.2 nM with an average of $7.2 \pm 4.4$ nM) were much higher than in polar waters along the transect ($0.8 \pm 2.0$ nM), and more in the range of the previously described wAW ($4.8 \pm 4.0$ nM) and fAW ($9.8 \pm 5.6$ nM). A similar increase was apparent for acetone, with values closer to wAW than to polar waters. DMS varied substantially (1.6 to 31.9 nM) at the sea-ice stations. DMS concentrations at the bloom stations 19 and 32 (see Sect. 2.1 and Fig. 4) were similar as in polar waters along the transect, whereas the pre-bloom stations (39, 43, 46) only showed > 2 nM DMS at 0.5 m depth. MeSH and CO both exhibited lower concentrations ($0.13 \pm 0.17$ nM and $1.45 \pm 1.67$ nM respectively) at the sea-ice stations. In contrast, isoprene concentrations were higher ($3.2 \pm 2.1$ pM) at the sea-ice stations compared to all other water masses along the transect. The substantial difference between polar waters observed in the northern part of the transect and surface values waters from sea-ice stations was also evident in Chl $a$, marker pigments of relevant phytoplankton groups (diatoms and *Phaeocystis*), and some trace gases (DMS, MeSH, CO, isoprene).

The shallow shelf stations 19 and 32 featured a marked phytoplankton bloom, with up to $10 \mu g L^{-1}$ of Chl $a$. The predominance of diatoms, constituting 90 % of phytoplankton, illustrates a typical spring bloom scenario (Degerlund and Eilertsen, 2010). Station 27, 31 and 47 had roughly 50 % diatom contribution while the pre-bloom stations 39, 43 and 46 were characterized by a mixed pico- and nanophytoplankton community of prasinophytes, chlorophytes, dinoflagellates, cryptophytes, chrysophytes and coccolithophorid-type haptophytes (Fig. S5) and Chl $a < 0.5 \mu g L^{-1}$. *Phaeocystis*, a typical bloom-forming organism in the high Arctic (Degerlund and Eilertsen, 2010), constituted up to 80 % of the phytoplankton biomass at station 47, but their biomass concentration was much lower compared to the *Phaeocystis* under-ice bloom found in the same region and year (Assmy et al., 2017). This indicates a declining bloom during our sampling period. Except for the Yermak Plateau stations (39, 43, 46), *Phaeocystis* contributed between 10 %–40 % of the phytoplankton biomass.

DMS and Chl $a$ were strongly correlated ($R^2$ Pearson's correlation coefficient = 0.93; Fig. S6). Isoprene also correlated with Chl $a$ ($R^2 = 0.6$, Fig. S6) but only when excluding station 19. This correlation supports a biological source of isoprene, in agreement with previous demonstrated links be-

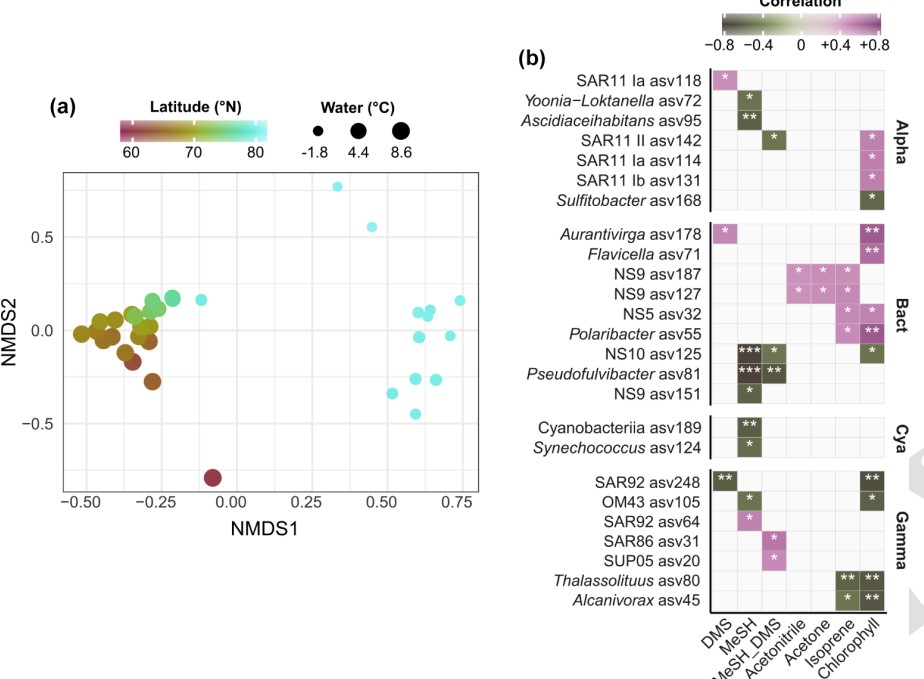

**Figure 3. (a)** Nonmetric multidimensional scaling of bacterial community composition (Bray–Curtis dissimilarities of Hellinger-transformed relative abundances). The colour gradient and dot size illustrate latitude and water temperature respectively. **(b)** Spearman correlations between environmental parameters and the abundance of bacterial ASVs. MeSH_DMS: ratio between MeSH and DMS, expressed as MeSH/(MeSH + DMS). Only correlations > |0.4| are shown, and only if stronger than with latitude. No correlations occurred with acetaldehyde and CO. Alpha: Alphaproteobacteria; Gamma: Gammaproteobacteria; Bact: Bacteroidetes; Cya: Cyanobacteria. Asterisks indicate Holm-corrected $p$ values (* < 0.05; ** < 0.01; *** < 0.001).

tween isoprene and Chl $a$ maxima (Tran et al., 2013). Station 19 was the only station where diatoms almost exclusively dominated the phytoplankton biomass. The little emission of isoprene by cold-water diatoms (Bonsang et al., 2010) could explain this pattern.

In contrast to the latitudinal transect, MeSH showed low concentrations at most ice stations, with the exception of station 19. Station 19 was special due to its location above the shelf and the diatom-dominated phytoplankton community. CO concentrations overall decreased with depth (Tran et al., 2013), except for station 31 with a CO peak at 30 m depth.

## 4 Discussion

### 4.1 Isoprene, CO, acetone, acetaldehyde and acetonitrile

Our study provides a comprehensive overview of biologically and climatically relevant trace gases in the microbiological context; covering ∼ 1400 nautical miles from 57 to 81° N, as well as under-ice vertical profiles north of Svalbard, in May–June 2015. Isoprene and CO concentrations can be compared with a previous study carried out in June–July 2010 (Tran et al., 2013; Table 1), i.e. 1 month later in summer and, hence, in a different water mass distribution but where only few phytoplankton blooms were encountered.

The concentrations of isoprene, usually associated with phytoplankton (Bonsang et al., 1992; Shaw et al., 2010), reported here were about 1 order of magnitude lower than described by Tran et al. (2013), even though the biomass indicator Chl $a$ was overall lower ($2\,\mu g\,L^{-1}$ compared to up to $8\,\mu g\,L^{-1}$ reported here). This may relate to seasonal differences in phytoplankton composition, as phytoplankton taxa are known to vary in their isoprene emissions (Bonsang et al., 2010; Shaw et al., 2010). Similar seasonal differences in isoprene concentrations were observed by Hackenberg et al. (2017), reporting on average 4.3 pM for March compared to 19.9 pM in July/August in the Arctic sector of the Pacific Ocean. Lower isoprene concentrations in polar waters correspond to Ooki et al. (2015), who found 27–33 pM in subpolar and transition waters and 4 pM in polar waters respectively. With regard to vertical profiles, the slight secondary maximum at 20–40 m depth may correspond to the Chl $a$ maximum, as reported by Tran et al. (2013). Nevertheless, the concentrations in our study are overall much lower (about 1 order of magnitude) than reported by Tran et al. (2013), indicating a high spatial variability of isoprene potentially re-

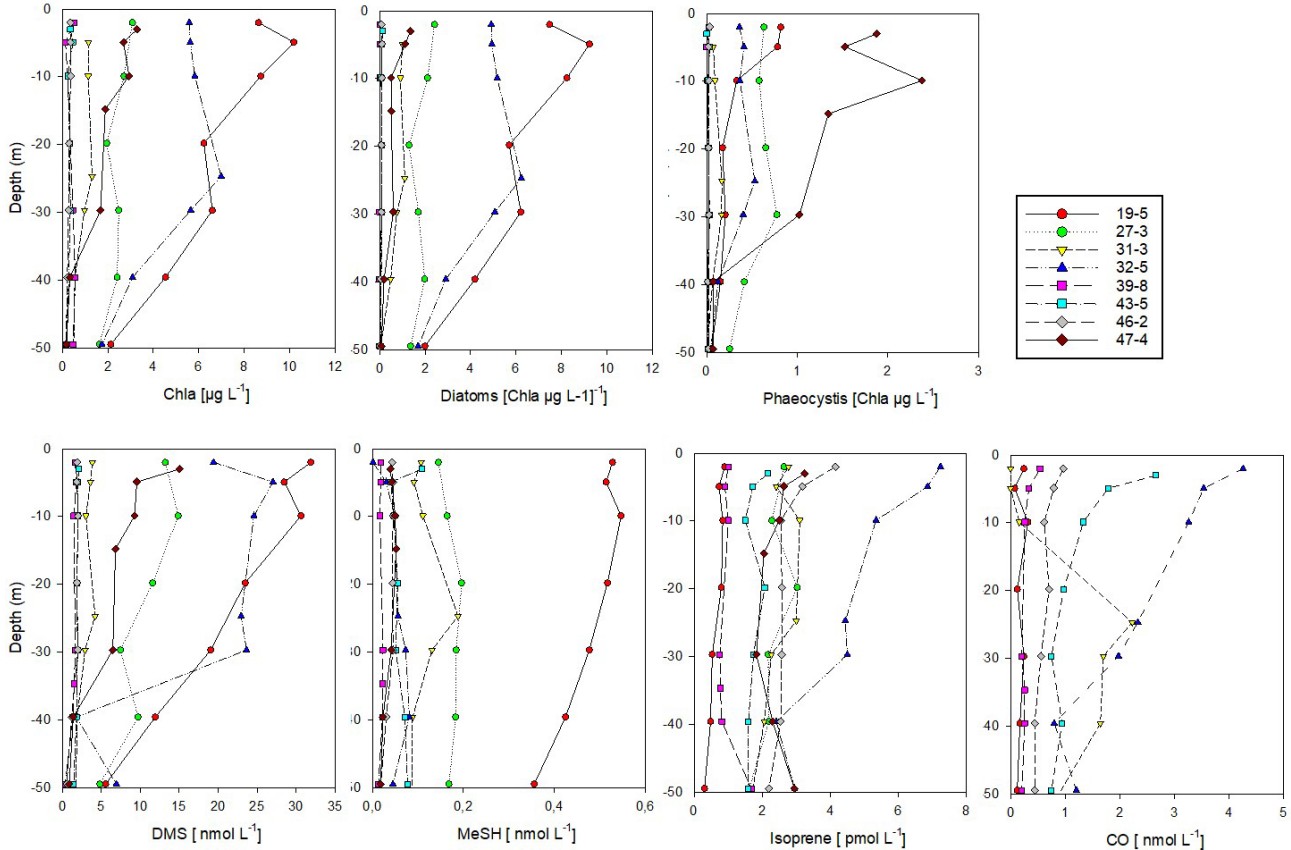

**Figure 4.** Vertical profiles of biological parameters and trace gas concentrations (0–50 m depth) at sea-ice-covered stations north of 80° N. According to Dybwad et al. (2021) stations 39, 43 and 46 (Yermak Plateau) were in a pre-bloom phase, while all other stations were in a bloom phase. Stations 19 and 32 were shelf stations. The contribution of each phytoplankton group is expressed as Chl *a* concentration. Data are stored under Peeken et al. (2023b).

lated to seasonally varying phytoplankton abundances. Simó et al. (2022) recently highlighted the importance of biological consumption of isoprene in seawater, possibly matching the magnitude of isoprene ventilated to the atmosphere, advising to consider both the sources and sinks when discussing isoprene concentrations and variability. The correlations of *Alcanivorax* and *Thalassolituus* ASVs with both isoprene and Chl *a* support phytoplankton as the source of this trace gas. *Alcanivorax* has been reported during phytoplankton blooms in the subarctic Atlantic (Thompson et al., 2020) and can degrade isoprene (Alvarez et al., 2009). *Alcanivorax* and *Thalassolituus* can be associated with microalgal surfaces and also perform hydrocarbon degradation (Love et al., 2021), indicating additional phytoplankton-linked factors that influence their distribution.

To date, CO in Arctic seawater has only been measured by Tran et al. (2013) in the same Arctic region, as well as by Xie and Zafiriou (2009) in the Beaufort Sea. The mean concentration of $5.9 \pm 2.9$ nM in polar waters along the transect matches previously reported averages ($6.5 \pm 3.2$ nM for Tran et al. (2013) and $4.7 \pm 2.4$ nM for Xie and Zafiriou (2009)

respectively). These concentrations are relatively high compared to the global oceanic mean of 2 nM CO (Conte et al., 2019). Elevated values in the Arctic are not reproduced by the NEMO-PISCES model (Conte et al., 2019), which might be caused by the bio-optical relationship between coloured dissolved organic matter (CDOM) and Chl *a*. NEMO-PISCES was originally developed for typical oceanic waters (Morel and Gentili, 2009). However, Arctic waters do not conform to this bio-optical type and are considered optically complex waters, with distinct signatures of CDOM and particle loads through the interplay of oceanic, riverine and ice-melt waters (Gonçalves-Araujo et al., 2018). Conte et al. (2019) attribute the release of CO and/or CDOM to sea-ice melt or to a lower bacterial consumption in cold waters. The first hypothesis is supported by up to 100 nM CO measured in sea ice (Xie and Gosselin, 2005; Song et al., 2011). Concerning CO in Atlantic waters, concentrations of up to $10.7 \pm 3.1$ nM are higher than in polar waters, and exceed data from the same region measured 5 years earlier (Tran et al., 2013) by up to 4 times. Highest CO values in temperate waters with low Chl *a* suggest that CO originated from an abiotic source,

e.g. the photodegradation of CDOM. In the ice-covered waters, the strong reduction with depth supports the notion that CO photoproduction decreases up to 3 times from the surface to 20 m depth (Fichot and Miller, 2010). However, in some vertical under-ice profiles, similar trajectories of Chl *a* and CO suggest an additional biological source, as shown by Tran et al. (2013). Biological sources of CO have been extensively studied by Gros et al. (2009), and the missing congruency at station 19 could be explained by the relatively low CO emission of cold-water diatoms (Gros et al., 2009), since diatoms accounted for the large bloom at the shelf station 19 (Fig. 3).

Considerable differences between Atlantic and polar waters occurred for acetone, acetonitrile and acetaldehyde. Reference data on acetone concentrations in seawater are scarce (Beale et al., 2013; Tanimoto et al., 2014; Wohl et al., 2020 and references therein), reporting 2 to 40 nM in the temperate and tropical Atlantic (Williams et al., 2004) as well as west Pacific oceans (Marandino, 2005). In the Atlantic Ocean, Yang et al. (2014) observed a mean value of 13.7 nM, without an obvious correlation to biological activity. For the Arctic Ocean, Yang et al. (2014) reported 6.8 nM in the Labrador Sea and Wohl et al. (2019) $8 \pm 2$ nM in the Canadian Arctic, matching our data between 65–70° N. Notably, these patterns match the southern Atlantic at 60° S (Wohl et al., 2020), suggesting similar dynamics in both subpolar regions. For acetone, the ocean is considered to be both a photochemical source and a microbial sink depending on the region (Jacob et al., 2002; Fischer et al., 2012). This dual role matches there herein observed relatively high values north of 60° N, and values down to the detection limit in polar zones. For acetonitrile, the oceans are a comparatively small source (originating from phytoplankton) or sink (through bacterial consumption) depending on location and season (see Singh, 2003; Williams et al., 2004, and references therein). The concentrations measured in the present study were mostly $> 1$ nM, and to our knowledge, the first reported in the Arctic Ocean. Overall, little is known about microbial utilization of acetone and acetonitrile, but biogenic effects have been suggested by Davie-Martin et al. (2020). Correlations of the NS9 clade (Flaobacteriales) with acetone and acetonitrile indicate an involvement in acetone and acetonitrile cycling among this diverse uncultured taxon.

Prior acetaldehyde measurements (Zhou and Mopper, 1997; Kameyama et al., 2010; Yang et al., 2014) reported 1.5 to 5 nM in the North Atlantic Ocean. In the present study, concentrations in AWs ($19.7 \pm 8.0$ nM) were on average 2-fold CE8 higher than those found in the North Atlantic by Yang et al. (2014) and Zhu and Kieber (2019). However, these related studies measured acetaldehyde in autumn, which could explain the difference, as the main source of acetaldehyde is attributed to photochemical degradation of CDOM.

## 4.2    DMS and MeSH

Previously reported DMS concentrations in polar oceans varied between 3 to 18 nM (Mungall et al., 2016; Jarníková et al., 2018; Uhlig et al., 2019), with up to 74 nM in the subsurface Chl *a* maximum in Baffin Bay (Galí et al., 2021). Hence, these are overall in the same range as our values. The average around 30 nM observed north of 80° N might be partly explained by the high DMS concentrations (up to 2000 nM) in sea ice (Levasseur, 2013; see also Hayashida et al., 2020). Indeed, ice-melt derived DMS can contribute up to 50 % to the water column inventory (Tison et al., 2010).

Stefels et al. (2007) suggested no direct relationship between DMS and Chl *a* on global scale, since the precursor of DMS (DMSP) is produced by diverse phytoplankton at different rates, connected to their physiological state. However, different approaches employed by Galí et al. (2018) and Wang et al. (2020) have shown that Chl *a* can be a strong predictor of DMS concentrations. In addition, Lana et al. (2012) CE10 reported that the DMS-Chl *a* correlation strongly varies with latitude, with a positive correlation at high latitudes (north of 40° N and south of 40° S). Nevertheless, we note that the figure presented by Lana et al. (2012) shows a lower correlation in the region covered by our transect. The missing correlation along our transect ($R^2 = 0.1$) likely reflects different phytoplankton types and bloom stages (Dybwad et al., 2021), whereas the strong correlation between Chl *a* and DMS in the Atlantic-influenced polar water masses at vertical under-ice profiles mirrored observations by Uhlig et al. (2019). Presumably, this is a typical marginal sea-ice zone effect, as found in other sectors of the Arctic Ocean (Galí and Simó, 2010; Levasseur, 2013; Park et al., 2013).

MeSH has seldom been quantified in marine waters to date, especially at polar latitudes. Leck and Rodhe (1991) reported on average 0.16 nM MeSH in the Baltic Sea and 0.28 nM and 0.34 nM in the North Sea respectively. Our data are 1 order of magnitude higher, ranging from $0.84 \pm 0.65$ nM in AWs to 3.49 nM in wPW, i.e. in the same range observed by Kiene et al. (2017) in the northeast subarctic Pacific Ocean. These authors showed that MeSH concentrations in surface waters generally decrease with depth, a feature which generally matches our under-ice vertical profiles although concentrations were low.

MeSH and DMS originate from the degradation of DMSP, mostly via bacterial demethylation (yielding MeSH) or cleavage (yielding DMS) (Moran and Durham, 2019; Lawson et al., 2020 and references therein). Laboratory experiments have indicated that the net yields of DMS and MeSH from DMSP are on average 32 % and 22 % respectively (Kiene, 1996). MeSH production might be promoted by low DMSP concentrations and high bacterial sulfur demand (Kilgour et al., 2022). Mesocosm experiments showed that the proportion of DMS versus MeSH increased from the prebloom phase to (induced) bloom conditions (Kilgour et al.,

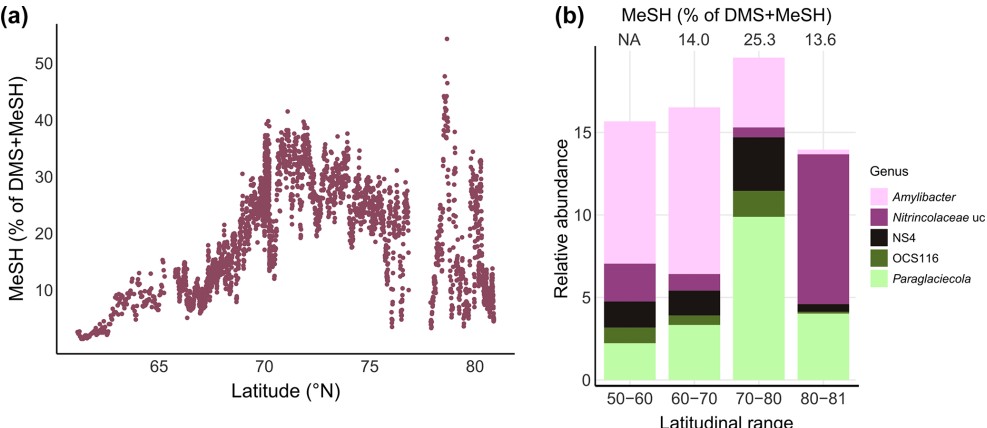

**Figure 5.** MeSH contribution to the sulfur budget and associated bacterial patterns. **(a)** Latitudinal variation of the MeSH fraction in relation to the total concentration of DMS + MeSH. **(b)** Relative abundance of selected bacterial genera by latitudinal range, , with the corresponding MeSH / DMS ratio displayed on top CE9. NA: not available; uc: unclassified. TS3

2022). In pelagic waters, DMS generally dominates gaseous sulfur, with MeSH being the second most abundant compound contributing on average $\leq 15\%$ to the total sulfur species in the North and Baltic seas (Leck and Rodhe, 1991), in the Atlantic Ocean (Kettle et al., 2001) and in the southwest Pacific Ocean (Lawson et al., 2020).

Comparable to some North Sea locations (Leck and Rodhe, 1991), MeSH contributed up to 40 % between 70–75° N in our study, with a maximum of 50 % at 78.6° N (Fig. 5a). This latitudinal variability was underlined by shifts in major bacterial genera. For instance, *Paraglaciecola* (Gammaproteobacteria) and NS4 (Bacteroidetes) peaked together with the highest MeSH fraction between 70–80° N. Abundances of *Amylibacter* decreased towards the north, whereas unclassified *Nitrincolaceae* prevailed north of 80° N together with an again smaller MeSH / DMS ratio (Fig. 5b). The overall MeSH contribution of 20 % suggests that MeSH represents a considerable fraction of sulfur, with linkages to microbial dynamics. Accordingly, we found several correlations with the abundance of specific ASVs. Correlations between *Yoonia-Loktanella* and *Ascidiaceihabitans* ASVs with MeSH reflect the prominent role of *Rhodobacteraceae* in DMSP demethylation (Curson et al., 2011; Moran et al., 2012). The positive relation of SAR11 and SUP05 ASVs corresponds to the prevalence of DMSP-metabolizing genes in these taxa (Nowinski et al., 2019; Landa et al., 2019; Sun et al., 2016). The link between cyanobacteria and MeSH potentially relates to the known uptake of DMSP by *Synechococcus* and *Prochlorococcus* (Vila-Costa et al., 2006), although DMSP-utilizing genes are overall rare in cyanobacteria (Liu et al., 2018). Overall, these observations indicate yet undescribed chemical linkages among primary producers.

## 5 Conclusion

We present the first measurements of DMS, MeSH and other trace gases along a transect from the North Atlantic to the ice-covered Arctic Ocean. High-resolution latitudinal data between 57 and 80° N were complemented with vertical profiles at sea-ice stations north of 80° N. Whereas isoprene, acetone, acetaldehyde and acetonitrile concentrations decreased northwards, CO, DMS and MeSH were uncorrelated with latitude and retained considerable concentrations in polar waters. Hence, these likely have phytoplankton-driven origins with regional variability, e.g. through localized blooms and/or the presence of sea ice. The DMS peak in polar waters pointed to sea ice as reservoir of DMS (Levasseur, 2013) and the prevalence of DMS-emitting phytoplankton. The marked correlation between DMS and Chl *a* in the diatom-dominated region north of 80° N represented a typical marginal sea-ice zone effect. The missing correlation between DMS and MeSH suggested different processes of production and degradation, although both compounds originate from DMSP. Although DMS was overall more abundant, MeSH contributed on average 20 % (and up to 50 %) to the total DMS + MeSH budget, suggesting consideration of MeSH as a secondary aerosol producer in some regions. The potential importance of MeSH was underlined by more and stronger bacterial correlations than with DMS, indicating that bacterial DMSP demethylation is important across extensive latitudinal gradients. Notably, higher acetaldehyde concentrations north of 80° N suggest that ice-covered regions could be a reservoir of acetaldehyde. While artefacts from off-line measurements (sampling through Niskin bottles) cannot be completely excluded, this result indicates a potential role of this reactive compound in regional atmospheric chemistry. However, a comprehensive understanding of marine trace gas dynamics, including the rapidly changing Arctic, requires further measurements in seawater, sea ice

and atmosphere. In conclusion, the reported trace gas concentrations in high spatial resolution provide important insights into climatically and biologically relevant compounds and their connection to microbiology.

*Code and data availability.* Data used in this study are listed below:

– Transect data: https://doi.org/10.1594/PANGAEA.953917 (Peeken et al., 2023a).

– Vertical profiles: https://doi.org/10.1594/PANGAEA.953908 (Peeken et al., 2023b).

– Sequence data: Raw sequence files have been deposited in the European Nucleotide Archive under accession number PR-JEB50492.

– Complete amplicon analysis workflow: https://doi.org/10.5281/zenodo.7609524 (Wietz et al., 2023; https://github.com/matthiaswietz/transsiz, last access: 6 February 2023).

*Supplement.* The supplement related to this article is available online at: https://doi.org/10.5194/bg-20-1-2023-supplement.TS4

*Author contributions.* VG, RSE, BB and IP designed the study. BB, VG and RSE performed trace gas measurements prior to the campaign; VG and RSE performed trace gas measurements on-board. IP coordinated all TRANSSIZ work, and supervised the biological sampling onboard as well as subsequent performed pigment analyses. AN coordinated the oceanographic sampling and water mass classification. KM set up the AUTOFIM sampling system and supervised DNA extraction. MW performed bacterial community analyses. VG, BB, IP and MW wrote the paper. All co-authors have read and contributed to the paper.

*Competing interests.* The contact author has declared that none of the authors has any competing interests.

*Acknowledgements.* We are thankful to the captain, crew and scientists from the TRANSSIZ expedition (ARK XXIX/1; PS92), carried out under grant number AWI_PS92_00. We thank Francois Truong and Jean-Eudes Petit for help with data processing, as well as Josephine Rapp and Halina Tegetmeyer for help with amplicon sequencing. We would like to thank the three anonymous reviewers who provided very useful comments that helped to improve the paper.

*Financial support.* Ilka Peeken, Matthias Wietz and Katja Metfies are funded by the PoF IV program "Changing Earth – Sustaining our Future" Topic 6.1 of the Helmholtz Association. The publication is part of the FRAM Observatory under EPIC number hdl:10013/epic.912d6e80-8865-431d-bc3b-a016f99166e9 TS5. Financial support was also provided by AWI, CNRS and CEA. CE11

*Review statement.* This paper was edited by Emilio Marañón and reviewed by three anonymous referees.

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

## Remarks from the language copy-editor

## Remarks from the typesetter