# Peer review of "Concentrations of dissolved dimethyl sulphide (DMS), methanethiol and other trace gases in context of microbial communities from the temperate Atlantic to the Arctic Ocean"

_Biogeosciences, 2022_

## Author Comment (AC1)

Comments from the reviewer are in black; answer to the reviewer are in blue, new adding to the text are in black and italics.

Rewiever 1

The paper by Gros et al. makes a substantial contribution to our understanding of the spatial distribution of climatically important trace gases and their potential underlying drivers. I would highlight the finding of very high DMS and significant MeSH concentrations under ice, the uncoupled DMS and MeSH distributions, and the distinct correlations between each VOC and bacterial ASVs at quite fine taxonomic resolution. Strong relationships between MeSH and bacterial ASVs in comparison to other VOCs is indicative of widespread bacterial DMSP metabolism. The paper is clearly written and well structured, and its messages are well supported by the observations. In the specific comments below I make some small criticisms that should be addressed. I suggest several additional citations, which I find important to both support the authors' findings and give fair credit to previous studies. I was also a bit disappointed by the little use authors make of HPLC data. My feeling is that they are missing an opportunity to assess the relative importance (even if based only on correlation patterns) of phytoplankton vs. heterotrophic bacterial diversity in controlling VOCs distribution in the area north of 80N that they sampled from Niskin bottles. My main criticism is with regards to the figures: they should be improved to make them more self-descriptive.

We thank the reviewer for the overall positive evaluation of our manuscript and for the useful comments and suggestions. We agree that it would have been great to link phytoplankton and bacterial diversity. However, please note that these data are unfortunately not fully comparable - HPLC & phytoplankton data are available for the depth profiles, while bacteria were only assessed in surface seawater. This prevents statistical comparisons. Please note, we have modified the figures according to your suggestions.

- Specific comments

- L39: Please add more up-to-date references, given that major advances in understanding of DMSP catabolism pathways have been made since 2007.
- L41: suggested citation: Kiene, 1996. Production of methanethiol from dimethylsulfoniopropionate in marine surface waters
- These articles may be of interest to provide a more complete view:
- L45-48: Rodríguez-Ros et al. 2020. Distribution and drivers of marine isoprene

- concentration across the Southern Ocean
- L49-50: Fichot and Miller, 2010. An approach to quantify depth-resolved marine photochemical fluxes using remote sensing: Application to carbon monoxide (CO) photoproduction
- L56: Acetaldehyde is also photoproduced: Zhu and Kieber, 2020. Global Model for Depth-Dependent Carbonyl Photochemical Production Rates in Seawater.
- L65: Lewis and Arrigo, 2020. Changes in phytoplankton concentration, not sea ice, now drive increased Arctic Ocean primary production
- L65: Galindo et al. 2014. Biological and physical processes influencing sea ice, under-ice algae, and dimethylsulfoniopropionate during spring in the Canadian archipelago
- L65: Wohl et al. 2022. Sea ice concentration impacts dissolved organic gases in the Canadian Arctic
- L66: Galí et al. 2019. Decadal increase in Arctic dimethylsulfide emission
- L68: Two good examples of changing phytoplankton species distribution
- Oziel et al. 2020. Faster Atlantic currents drive poleward expansion of temperate phytoplankton in the Arctic Ocean
- Orkney et al. 2020. Bio-optical evidence for increasing Phaeocystis dominance in the Barents Sea

  We thank the reviewer for these suggestions of references, which have all been included in the introduction. In addition, we now also mention the new review on DMS including polar data https://doi.org/10.3390/microorganisms10081581 as well as the review of the microbiology of isoprene in aquatic system (https://doi.org/10.3354/ame01972)

- L259-260: I don't see how the correlation between DMS and Chl is connected to diatoms being the main photosynthetic group. Please rephrase.

- We apologize for this phrase, which indeed is misleading. The new text reads: "*We observed a strong correlation between DMS and Chl a (R-squared Pearson's correlation coefficient = 0.93; Fig. S8). Since diatoms were the most prominent photosynthetic group at ice-covered stations (Fig. 4) we consider them important for DMS fluxes in the Polar Ocean.*"

- L261: I cannot see the cyan squares of station 43 in the CO panel.

  Thank you for spotting this, we had mixed up the columns for CO when transferring it from Excel to Sigmaplot for the final version. Please find a new Fig. 4 with the correct CO profiles, which were not carried out at every station.

[Figure]

***Fig. 4.*** *Biological parameters and trace gas vertical distribution (0-50 m depth) at sea-ice covered stations north of 80°. According to Dybwad et al., (2021) stations 39, 43 and 46 (Yermak Plateau) were in a pre-bloom phase, while all other stations were in a bloom phase. Stations 19 and 32 were shelf stations. The contribution of each phytoplankton group is expressed as Chl a concentrations.*

- L305: Perhaps mention that Arctic waters feature much higher CDOM content than typical oceanic waters. The approach used by Conte et al. (2019) estimated CDOM using a biooptical relationship between CDOM and Chl developed for typical (case I) oceanic waters (Morel et al., 2009). Arctic waters do not conform to this bio-optical type and are typically seen as optically complex waters with compound influence of oceanic, riverine and icemelt waters with distinct signatures in terms of CDOM and particle loads. Failing to account for the high CDOM content is likely to result in underestimation of CO photoproduction.

We thank the reviewer for this clarification and changed the text accordingly:

"*Elevated values in the Arctic are not reproduced by the NEMO-PISCES model (Conte et al., 2019), which might be caused by the bio-optical relationship between CDOM and Chl-a. This model was originally developed for typical oceanic waters (Morel & Gentilli., 2009). However, Arctic waters do not conform to this bio-optical type and are typically considered optically complex waters, with distinct signatures of CDOM and particle loads through the influence of oceanic, riverine and ice-melt waters (Goncalves-Araujo et al. 2018). Conte et*

*al., 2019 attribute the release of CO and/or CDOM to sea-ice melt or to lower bacterial consumption in cold waters. The first hypothesis is supported by up to 100 nM CO measured in sea ice (Xie and Gosselin, 2005; Song et al., 2011)."*

- L332: A note of caution: the values reported by Davie-Martin et al. (2020) for the NAAMES expedition are not credible in the case of DMS production rates. The highest DMS production they found in May, around 43 nM h-1 (1000 nM d-1), is about 15 times higher than any previous measurement (Galí and Simó, 2015, their figure 3a). This might have been caused by the bubbling in their experimental setup, which is known to induce DMS production in stressed cells (eg Wolfe et al., 2002. Dimethylsulfoniopropionate cleavage by marine phytoplankton in response to mechanical, chemical, or dark stress). Similar artifacts may have affected measurements of the production rate of other VOCs in that study.

  We thank the reviewer for this note of caution. Concerning the statement about acetonitrile for which oceans can be a small source or sink, we have replaced the citation of Davie-Martin et al. (2020) by Singh et al. (2003) and Williams et al. (2004). We have nevertheless left this citation to underline the potential microbial utilization of acetone and acetonitrile.

- L349: Suggested citation: Hayashida et al. 2020. Spatiotemporal Variability in Modeled Bottom Ice and Sea Surface Dimethylsulfide Concentrations and Fluxes in the Arctic During 1979 – 2015. Quoting from their abstract: "...model results indicate that the bottom ice DMS and its precursor dimethylsulfoniopropionate production can be the only local source of oceanic DMS emissions into the atmosphere during May prior to pelagic blooms".

  The citation of Hayashida has been added, in addition to Levasseur, 2013.

- L350-351: this view can be nuanced. At high latitudes, the seasonal correlation between DMS and Chl is typically positive and quite high. See e.g. Lana et al. (2012) Reexamination of global emerging patterns of ocean DMS concentration, their fig. 4. Also:

- Galí et al., 2018. Sea-surface dimethylsulfide (DMS) concentration from satellite data at global and regional scales. Table 1 and Fig. 7.
- Wang et al., 2020. Global ocean dimethyl sulfide climatology estimated from observations and an artificial neural network. Table 1, section 3.

We thank the reviewer for this comment. We have nuanced this view in the revised version, now reading as follow :

*"Stefels et al. (2007) have suggested no direct relationship between DMS and Chl a on global scale, since the precursor of DMS (DMSP) is produced by diverse phytoplankton at different rates, connected to their physiological state. However, different approaches employed by Gali et al. (2018) and Wang et al. (2020) have shown that Chl a can be a strong predictor of DMS concentrations. In addition, Lana et al. (2012) reported that the DMS-Chl a correlation strongly varies with latitude, with a positive correlation at high latitudes (north of 40°N and south of 40°S). Nevertheless, we note that the figure presented by Lana et al. (2012) shows a lower correlation on the region covered by our transect. The poor correlation found along our transect ($R^2$ =0.1) probably reflects different phytoplankton types and bloom stages.*

- It would be very helpful to see the main currents and the different water masses on this map. For example, this would support the description given in L180-186, several parts of the Results and Discussion, the summary given in Table 1, etc.

We thank the reviewer for this suggestion, which was also suggested by reviewer 2. We have modified Figure1 (and caption) to now include the main surface currents (arrows) as well as the water masses (colored dots) along the ship track (the latter was earlier shown in Fig S6). It also includes a legend for the bathymetry (background blue colors). The new figure is perhaps a little busy, but has the benefit of including all relevant information in one figure, better supporting several parts of the manuscript. Fig S6 has thus been omitted.

[Figure]

*NEW Fig. 1: The ship track with the sampled range colored by water mass: 'regular' warm Atlantic Water (wAW), coastal influenced Atlantic water with low salinity (AWs), freshened and cooled Atlantic Water (fAW), warm Polar Water (wPW) and cold Polar Water (wPW), according to the temperature and salinity criteria in Table 1. Surface measurements were sampled by the FerryBox system between 57°N and 81°N, while vertical profiles were sampled by a CTD rosette at eight sea ice stations (black insert and Table S1). The background map shows the GEBCO_2022 bathymetry (GEBCO Compilation Group (2022); doi:10.5285/e0f0bb80-ab44-2739-e053-6c86abc0289c) and a schematic overview of the major currents influencing the surface waters in the study area, as adopted from Skagseth et al. (2022): the northward flowing Norwegian Atlantic Slope Current (NwASC), West Spitsbergen Current (WSC), Norwegian Atlantic Front Current (NwAFC), and Norwegian Coastal Current (NCC) and the southward East Greenland Current (EGC).*

The current-related text in the manuscript (section3.1) was also slightly modified with the aim to better describe the sampled water masses along the ship track:

*"Along the latitudinal transect, we performed online surface measurements of Chl a and hydrographic parameters, covering five different water masses: warm Atlantic Water with low salinity (AWs), 'regular' warm Atlantic Water (wAW), freshened and cooled Atlantic Water (fAW), cold Polar Water (cPW) and warm Polar Water (wPW) as defined in Table 1 following Tran et al. (2013). The major part of the*

*transect, from 63 to 80°N, occurred in wAW (see Fig. 1). Fresher (low-saline) Atlantic Water (AWs and fAW) was encountered in the vicinity of the Norwegian Coastal Current (NCC) which carries water masses influenced by river run-off: AWs at 60.6-62.3 °N (with fAW in the mixing zones), and fAW around 70-72°N. Fresher mixed products (fAW) were also intermittently encountered west of Svalbard (where AW meets fjord/coastal water masses), as well as in the marginal ice zone where AW mixes with and gradually subducts under fPW. Polar Water (PW) only occurred north of 80°N in the Nansen Basin."*

- Figure 2: I strongly recommend to depict somehow the water masses along the transect, for example with colored horizontal bars on top of the plot.

We thank the reviewer for this idea, which was also suggested by reviewer 2. The revised figure includes now as a first panel a horizontal bar representing the different water masses.

[Figure]

**Figure 2:** Latitudinal variability of acetone (nM), acetaldehyde (nM), acetonitrile (nM), isoprene (pM), DMS (nM), MeSH (nM), and CO (nM) between 57.2°N to 80.9°N in relation to Chl a (µg L⁻¹) and water temperature (°C). Due to sensor failure temperature values are missing until ~61°N. The top X-scale gives the corresponding date. On the top panel, the colored horizontal bar represents the different water masses as in Figure1 and Table 1

- Figure 3: It would be useful to provide more commonly used taxonomic classifications/levels for some bacterial genera. For example, Yoonia-Loktanella and Ascidiaceihabitans tell nothing to me, but I immediately associate Rhodobacteraceae with certain types of reduced sulfur metabolism.

In the discussion, Yoonia-Loktanella and Ascidiaceihabitans are specified as Rhodobacteraceae and how this family commonly performs sulfur cycling. Rhodobacteraceae are also mentioned in the abstract. We think that this figure would become difficult to read if more taxonomic info were added to each ASV, and would therefore prefer to leave it as is.

- Figure 4: Please indicate (for example in the legend) whether stations are in pre-bloom or bloom stage, perhaps distinguishing the shelf stations as well.

Thank you for this suggestion, the new figure captions reads:

*"**Fig. 4.** Biological parameters and trace gas vertical distribution (0-50 m depth) at sea-ice covered stations north of 80°. According to Dybwad et al., (2021) stations 39, 43 and 46 (Yermak Plateau) were in a pre-bloom phase, while all other stations were in a bloom phase. Stations 19 and 32 were shelf stations. The contribution of each phytoplankton group is expressed as Chl a concentrations."*

- Figure 5: same as Fig. 2.

As for Fig. 2, we have tested plotting the horizontal bar with the water masses classification, but found that it does not add meaningful information. Hence, we have decided to leave the figure as is.

- Technical corrections and typos
- L156: "at their" repeated
- L254: "but the here found concentrations" sounds a bit awkward, please reword
- L292: "as already been found", please remove "been"
- L324: nM, not nm

All technical corrections have been done

---

## Author Comment (AC2)

Comments from the reviewer are in black; answer to the reviewer are in blue, new adding to the text are in black and italics.

Reviewer 2

General Comments

This manuscript describes a comprehensive set of dissolved reactive gas and microbiological measurements taken on a research cruise from the North Atlantic to the Arctic. The authors extensively discuss the sources and relationships between the measured trace gases and microbiology, and focus their discussion on dimethyl sulfide (DMS) and methanethiol (MeSH). They show that MeSH does not correlate with DMS during the entirety of the cruise. They find that MeSH can contribute on average 20%, and up to 50%, to the total waterside sulfur budget, defined by the sum of DMS and MeSH. Overall, this manuscript is well-structured and presents new findings that are valuable to the biogeosciences and atmospheric chemistry communities and should be published after the following main comments are addressed. My main comment for this manuscript is that a more nuanced discussion of variations in the measured MeSH/(DMS+MeSH) ratio and the dominant factors controlling it would be extremely helpful. Little information currently exists on how this ratio varies based on environmental parameters, which has impacts for how we think about $SO_2$ production.

This dataset provides measurements of MeSH in a region for the first time with varying temperature and salinity, meaning that we now have data to form more accurate models of $SO_2$ production based on how this ratio scales with different waterside parameters.

We thank the reviewer for the overall positive evaluation of our manuscript and for the useful comments and suggestions. We address the different points in the sections below.

Suggestions for discussion on this topic include:

Addition of a column containing MeSH/(DMS+MeSH) to Table 1. Can any trends from the water classifications (salinity, temperature) explain the observed variations?

In line 371-372, it's noted that at 78.6 ⁰N, MeSH contributes up to 50% of total sulfur, but only 20-40% in 70-75ºN. What is driving this difference? all this needs more discussion, check in detail

Looking at Fig. 2, it looks like in some regions MeSH and DMS covary (>71ºN) and in some regions, there is little correlation (<68ºN). Some more statistical analysis and discussion of why there seems to be a correlation in certain water masses/time periods but not others would be useful.

Fig 5 could be revised to provide more information about any environmental parameters controlling this ratio (colored points by salinity, temperature, chlorophyll?) or additional regressions against these variables instead of just latitude.

We thank the reviewer for the suggestions how to improve the discussion on MeSH/(MeSH+DMS). We have added the MeSH / (MeSH +DMS) ratio to Table 1 (see below), showing that this ratio was higher in Atlantic (wAW and fAW) than in Polar waters. However, we did not find any significant global correlation between the ratio and environmental parameters (chl-a, temperature). We have tested coloring the dots in Fig.5 by environmental parameters, but found it provided little additional information. Nevertheless, to highlight the latitudinal variability and possible underlying links, we have added Fig. 5b showing major bacterial genera that vary by latitudinal ranges and their specific MeSH / (MeSH +DMS) ratios. In addition to the figure, we have added the following text (new text in italics and bold):

Comparable to some North Sea locations (Leck and Rodhe, 1991), MeSH contributed up to 40% between 70°N-75°N, with a maximum of 50% at 78.6°N (Fig. 5a). ***This latitudinal variability*** was underlined by shifts in major bacterial genera. ***For instance,*** **Paraglaciecola (Gammaproteobacteria), NS4 (Bacteroidetes) peaked together with the** *highest* **MeSH fraction between 70-80°N. Amylibacter** ***decreased towards the north, whereas*** ***unclassified*** **Nitrincolaceae** ***prevailed >80°N together with an again smaller MeSH/DMS ratio.*** The overall MeSH contribution of 20% suggests that MeSH represents a considerable fraction of sulphur, being linked to microbial dynamics. Accordingly, we found several correlations with the abundance of specific ASVs. Correlations between *Yoonia-Loktanella* and *Ascidiaceihabitans* ASVs with MeSH reflected the prominent role of *Rhodobacteraceae* in DMSP demethylation (Curson et al., 2011; Moran et al., 2012). The positive link of SAR11 and SUP05 ASVs corresponds to the prevalence of DMSP-metabolizing genes in these taxa (Nowinski et al., 2019; Landa et al., 2019; Sun et al., 2016). The link between cyanobacteria and MeSH was notable, since DMSP-utilizing genes appear to be rare in cyanobacteria (Liu et al., 2018). Hence, there might be indirect effects on other photosynthetic organisms, indicating yet undescribed chemical linkages among primary producers.

|  | Acetonitrile (nM) | Acetaldehyde (nM) | Acetone (nM) | DMS (nM) | Methanethiol (nM) MeSH/(MeSH +DMS) | Isoprene (pM) | CO (nM) |
|---|---|---|---|---|---|---|---|
| Coastal-influenced/low-salinity Atlantic Water (AWs; θ>5°C, S<34.4) | 1.1 ± 0.6 | 19.7 ± 8.0 | 23.3 ± 12.8 | 15.7 ± 7.0 | 0.8 ± 0.7 5.6 ± 7.1 | 2.6 ± 0.8 *23.4 ± 3.10** | 10.7 ± 3.1 *2.50 ± 1.70** |
| warm Atlantic Water (wAW; θ>2°C, S>34.9) | 0.5 ± 0.2 | 4.8 ± 4.0 | 2.4 ± 5.9 | 11.8 ± 7.0 | 2.9 ± 1.5 21.9 ± 8.7 | 1.4 ± 0.7 *42.5 ± 49.6** | 5.9 ± 2.8 *3.3± 2.2** |
| freshened Atlantic Water (fAW; θ>1°C, 34.4<S<34.9) | 0.9 ± 0.4 | 9.8 ± 5.6 | 14.6 ± 10.8 | 13.1 ± 8.8 | 3.3 ± 1.5 20.7 ± 10.6 | 2.7 ± 1.5 *24.8. ± 19.1** | 10.2 ± 5.9 *3.4 ± 2.4** |
| cold Polar Water (cPW; θ<0°C, S<34.7) | 0.3 ± 0.1 | 1.0 ± 2.3 | BDL | 30.0 ± 9.3 | 2.8 ± 0.8 9.1 ± 2.3 | 1.2 ± 0.5 | 5.0 ± 2.8 |
| warm Polar Water (wPW; θ>0°C, S<34.4) | 0.2 ± 0.1 | 0.3 ± 0.9 | BDL | 34.7 ± 8.5 | 3.5 ± 0.3 9.6 ± 2.0 | 1.1 ± 0.3 | 7.8 ± 2.1 |
| Polar waters (PW) (cold+warm) | 0.3 ± 0.1 | 0.8 ± 2.1 | BDL | 31.2 ± 9.3 | 3.0 ± 0.7 9.2 ± 2.2 | 1.2 ± 0.4 *14.5 ± 11.5** | 5.9 ± 2.9 *6.5 ± 3.2** |
| Surface water at sea-ice stations > 80°N (range) | 0.3 ± 0.1 (0.2-0.5) | 7.2±4.4 (0.3-14.2) | 2.3±2.8 (0-6.9) | 11.2 ±10.9 (1.6-31.9) | 0.1±0.2 (0.02-0.5) | 3.2 ± 2.1 (0.9-7.3) | 1.5 ± 1.7 (0.2-4.3) |

*New Table 1*

[Figure]

*New Fig. 5, with added panel Fig. 5b: Relative abundance of selected bacterial genera by latitudinal range and associated MeSH/DMS ratio. NA: not available; uc: unclassified.*

Specific Comments – Manuscript

Line 19: It would be helpful to make it clear somewhere in the abstract that all gas measurements are in the dissolved phase in the seawater and not in the air. Potentially could also add "dissolved" to title.

Dissolved has been added to the title (and "concentrations" replaced by "variability" to answer a suggestion of reviewer 3). The new title reads:

*"Concentrations of dissolved dimethyl sulphide (DMS), methanethiol and other trace gases in context of microbial communities from the temperate Atlantic to the Arctic Ocean".*

Line 39: Instead of "rapidly oxidized", can you state the atmospheric lifetime of DMS?

Has been changed to:

*"DMS is rapidly oxidized once emitted to the atmosphere (average lifetime of 1 day)"*

Line 40: It would be useful to add some references to the CLAW hypothesis. Some

suggestions:

Bates, T. S., Lamb, B. K., Guenther, A., Dignon, J., and Stoiber, R. E.: Sulfur emissions to the atmosphere from natural sourees, J. Atmos. Chem., 14, 315–337,

https://doi.org/10.1007/BF00115242, 1992.

Charlson, R. J., Lovelock, J. E., Andreae, M. O., and Warren, S. G.: Oceanicphytoplankton, atmospheric sulphur, cloud albedo and climate, Nature, 326, 655–661,

https://doi.org/10.1038/326655a0, 1987.

Line 41: Suggested references for DMSP demethylation producing MeSH:

Kiene, R. P.: Production of methanethiol from dimethylsulfoniopropionate in marine surface waters, Mar. Chem., 54

Kiene, R. P. and Linn, L. J.: The fate of dissolved dimethylsulfoniopropionate (DMSP) in seawater: tracer studies using 35S-DMSP, Geochim. Cosmochim. Ac., 64, 2797–2810,https://doi.org/10.1016/S0016-7037(00)00399-9, 2000b.

Line 42: The atmospheric impacts of MeSH are less well-characterized than DMS, but we do know some about MeSH impacts based on its oxidation and reactivity. See references

below:

Butkovskaya, N. I. and Setser, D. W.: Product Branching Fractions and Kinetic Isotope

Effects for the Reactions of OH and OD Radicals with CH3SH and CH3SD, J. Phys. Chem. A, 103, 6921– 6929, https://doi.org/10.1021/jp9914828, 1999.

Tyndall, G. S. and Ravishankara, A. R.: Atmospheric oxidation of reduced sulfur species, Int. J. Chem. Kinet., 23, 483–527, https://doi.org/10.1002/kin.550230604, 1991

Novak, G. A.; Kilgour, D. B.; Jernigan, C. M.; Vermeuel, M. P.; Bertram, T. H. Oceanic Emissions of Dimethyl Sulfide and Methanethiol and Their Contribution to Sulfur Dioxide Production in the Marine Atmosphere. Atmospheric Chem. Phys. 2022, 22 (9), 6309–6325. https://doi.org/10.5194/acp-22-6309-2022.

Line 49: Isoprene has also been shown to have a photochemical source.

Ciuraru, R.; Fine, L.; Pinxteren, M. van; D'Anna, B.; Herrmann, H.; George, C. Unravelling New Processes at Interfaces: Photochemical Isoprene Production at the Sea Surface. Sci. Technol. 2015, 49 (22), 13199–13205. https://doi.org/10.1021/acs.est.5b02388.

Line 52-53: Add a citation for OVOCs affecting the oxidative capacity of the remote atmosphere. Potentially this one could work:

Singh et al. (2004). Analysis of the atmospheric distribution, sources, and sinks of oxygenated volatile organic chemicals based on measurements over the Pacific during TRACE-P. Journal of Geophysical Research Atmospheres.

We thank the reviewer for the suggested references, which have been included in the text.

Lines 55-59: Acetone, methanol, acetonitrile, and acetaldehyde can also be anthropogenic, affecting whether the net flux is positive or negative. It is worth adding this in addition to whether the flux is positive or negative depending on oligotrophic water.

Added

Lines 58-59: Another reference for acetone and acetaldehyde flux.

Phillips, D. P., Hopkins, F. E., Bell, T. G., Liss, P. S., Nightingale, P. D., Reeves, C. E., Wohl, C., an d Yang, M.: Air–sea exchange of acetone, acetaldehyde, DMS and isoprene at a UK coastal site, Atmospheric Chem. Phys., 21, 10111–10132, https://doi.org/10.5194/acp-21-10111-2021, 2021.

Additional reference was added

Line 115: What do the different blue colors mean in Fig. 1? I suggest adding information about the water classifications to this figure as well, like Fig S6.

We thank the reviewer for this suggestion, which was also suggested by reviewer 1. We have modified Figure1 to now include the main surface currents (arrows) as well as the water masses (colored dots) along the ship track (the latter was earlier shown in Fig S6). It also includes a legend for the bathymetry (background blue colors). The new figure is perhaps a little busy but has the benefit of including all relevant information in one and the same figure, as better support to several parts of the manuscript. Fig S6 has thus been omitted.

[Figure]

*New figure 1:* *The campaign ship track with the sampled range colored by water mass: 'regular' warm Atlantic Water (wAW), coastal influenced Atlantic water with low salinity (Aws), fresh Atlantic Water (fAW), warm Polar Water (wPW) and cold Polar Water (wPW), according to the temperature and salinity criteria in Table 1. Surface measurements were sampled by the FerryBox system between 57°N and 81°N while vertical profiles were sampled at eight sea ice stations (black insert and Table S1). The background map shows the GEBCO_2022 bathymetry (GEBCO Compilation Group (2022); doi:10.5285/e0f0bb80-ab44-2739-e053-6c86abc0289c) and a schematic overview of the major currents influencing the surface waters in the study area, as adopted from Skagseth et al. (2022): the northward flowing Norwegian Atlantic Slope Current (NwASC), West Spitsbergen Current (WSC), Norwegian Atlantic Front Current (NwAFC), and Norwegian Coastal Current (NCC) and the southward East Greenland Current (EGC).*

Line 152: It has been shown previously that other molecules can be measured in PTR-MS at the unit mass 63 where DMS is measured, such as ethylene glycol. Has this been accounted for in background measurements? Otherwise, if these are measured along this transect, they could artificially inflate the DMS measurements. It should be explicitly stated that all VOC measurements were taken at their unit mass m/z + 1 mass.

The m/z +1 has been attributed uniquely to DMS (like other studies using a HS-PTRMS in oceanic environments, see for example Wohl et al. 2020, https://doi.org/10.5194/bg-17-2593-2020). It is indeed very likely that compounds like ethylene glycol have negligible interference in these oceanic environments. In addition, the strong correlation (R2 = 0.93) observed for m/z 63 with Chl-a on the vertical profiles confirms that m/z 63 can be attributed to DMS only. The text already states that the compounds are measured at their m/z +1.

Line 155: Cite Blake et al. (2009) again for the thermodynamics of the proton transfer reaction.

Citation has been added.

Line 155-156: While PTR-MS can be a soft ionization technique, there is still the possibility for fragmentation of larger molecules to affect your measurements, so quantifying at the m/z +1 mass may be the protonated molecule in addition to fragments of larger molecules. Have there been control experiments to support quantifying the molecules of interest only at the m/z+1 mass?

The reviewer is right to mention the possibility for fragmentation. However, for the masses reported in this paper, the fragmentation of larger molecules are likely not impacting significantly the measured m/z+1 of interest. Nevertheless, this is an important point to mention and we have added the following precisions to the revised manuscript (section 2.3.1)

*"The soft ionization allowing only small fragmentation, the compounds are directly measured at their corresponding m/z+1. Although we cannot rule out higher molecule fragment on the measured m/z+1, interferences from other compounds are likely negligible for the masses presented in this manuscript (Blake et al., 2009; Yuan et al. (2017). An exception could be isoprene, as it can contain fragmentation of 2-methyl-3-buten-2-ol (MBO) or fragmentation of cyclohexanes. However, during the period when the PTRMS was measuring in scan mode, MBO mass (m/z 87) has shown no correlation (R2= 0,02) with m/z 69. In addition, the good correlation (R2= 0.77) between isoprene and Chl a on vertical profiles (see Figure S7) confirms that the measured m/z 69 can be mainly attributed to isoprene. For acetone, the signal corresponds to "acetone + propanal" but it has been noted that propanal can be neglected and m/z 59 be considered as acetone (de Gouw and Warneke, 2007). The PTRMS used for this campaign had been used the year before on a field campaign and some of its characteristics are described here (Zannoni et al., 2016)."*

Line 157-158: What is the residence time in your tubing? Does it affect the measurements of any of your molecules, like acetonitrile?

Thank you for mentioning this. We explain it now in the text as followed:

*The estimated residence time of about 30 seconds should prevent any degradation or adsorption of the extracted gases in the system. Furthermore, a series of standards were measured under the same experimental conditions, showing high linearity in the system's response. This observation supports the absence of artefacts in the experimental procedure.*

Line 164-164: I don't see information in the SI on how MeSH was calibrated. Was this an assumed equivalent sensitivity as DMS? This calibration should also be included in the SI.

The sensitivity of MeSH (m/z 49) has been determined by taking an average sensitivity factor (13.4 ncps/ppb) between the sensitivity from the 2 "surrounding" compounds (m/z 45, acetaldehyde and m/z 59, acetone) with similar sensitivity (within 6%) (13.0 ncps/ ppb and 13.8 ncps/ppb respectively).

This point is now included in the SI, as well as the fact that it represents an additional uncertainty for MeSH concentrations. Nevertheless, the estimated sensitivity for MeSH corresponds to the high range of the determined sensitivity coefficients. Hence, the coefficient could be over-estimated, and which would even result in underestimating the corresponding MeSH concentrations. Thus, we feel very confident in our conclusion that MeSH plays a significant role in the sulphur budget, since we rather under- than overestimating this compound with our method.

Line 194: I don't see any discussion of uncertainty for the trace gas measurements. Some discussion of this should be included either in the methods or in 3.1.1.

We thank the reviewer for pointing this out. The following information has been added in the method description.

*The measurement uncertainty with this PTRMS had been estimated at ± 20% taking into account errors on standard gas, calibrations, blanks, reproductibility/repeatability and linearity (Baudic et al., 2016). The overall uncertainty for dissolved VOC was estimated at ± 30%.*

Line 195: What are your detection limits for acetone and acetaldehyde?

We have added in the supplementary S4 the following information

*During the campaign, a blank of the system was determined by injecting only the extraction gas through the system, taking into account the instrumental background noise from the instrument and potentially residual VOCs in the extraction gas. This value was subtracted from the measurements. The detection limit was estimated as 3 sigma of the blank variability, variying from 0.3 nM (for acetonitrile) to 3 nM (for acetone and acetaldehyde). Some values shown in Fig. 2 are below the estimated detection limit for acetone and acetaldehyde; which is due to the subtraction of the blank (the measured signal was above the detection limit). Values have been kept in the figure to show the variability, but a note has been added in the figure caption ("Values below 3 nM are below the detection limit for acetone and acetaldehyde, see S4).*

Line 205: I'm curious what's causing the high MeSH between 70 and 73-75 °N?

This could be due to difference in bacteria community. This is now discussed with the new figure 5b (see above).

Line 209: I think this figure can be edited to help the story flow better. My suggestions are:

Since CO presumably has a different source than DMS and MeSH, having it on the same panel is distracting. I'd suggest making this a 4-panel figure with CO on its own.

Can some information about the water masses (info from Table 4) be included? Perhaps as a shaded background.

Can information about the timing of these measurements be included? By plotting against latitude, it is hard to understand how many points are represented at each latitude, especially in the horizontal transect region near 70°N.

We thank the reviewer for all suggestions. The revised figure 2 (see below) now includes a horizontal bar representing the different water masses, as suggested by reviewer 1 too. We have furthermore added an additional X-scale to highlight the timing of the measurements.

We have decided to keep CO on the same panel, for two reasons. As we have already added a bar at the top, adding another panel would limit readability of the figure. Moreover, having CO on the same panel as DMS and MeSH allows to easily visualizing some common features (for example higher values between 73°N and 75°N).

[Figure]

**Figure 2:** Latitudinal variability of acetone (nM), acetaldehyde (nM), acetonitrile (nM), isoprene (pM), DMS (nM), MeSH (nM), and CO (nM) between 57.2°N to 80.9°N in relation to Chl a (µg L$^{-1}$) and water temperature (°C). Due to sensor failure temperature values are missing until ~61°N. The top X-scale gives the corresponding date. On the top panel, the colored horizontal bar represents the different water masses: warm Atlantic Water with low salinity (AWs), 'regular' warm Atlantic Water (wAW), freshened and cooled Atlantic Water (fAW), cold Polar Water (cPW) and warm Polar Water (wPW).

Line 227: I am unclear what MeSH_DMS is on Fig. 3b x-axis? Is this MeSH/(MeSH+DMS)?

If so, should be updated to read more clearly.

The reviewer is correct. The figure legend has been clarified accordingly:

*"MeSH_DMS: ratio between MeSH and DMS, expressed as MeSH/(MeSH+DMS)"*

Technical Corrections – Manuscript

Line 35: "source and sink" should be "sources and sinks"

Line 99: "some leads present )," should be "some leads present),"

Line 123: "Chl a l concentrations" should be "Chl a concentrations"

Line 156: "at their at their" should be "at their"

Line 254: "but the here found concentrations" should be "but here the concentrations found"

Line 324: "nm" should be "nM"

All technical corrections have been done.

Technical Corrections – Supplemental

Line 8: There is something cut off in the upper righthand corner of Fig. S1.

This was due to a line number partly covering the figure, this does not appear in the version without the line numbering.

Line 61: Fig. S3 is blurry and hard to read.

A clearer version of the figure is now included

[Figure]

Line 77-79 : I am unclear what it means for the Henry' law constant "whatever the solubility of the compound over 4 to 5 orders of magnitude"

We apologize; there was a problem in the X and Y axis units, which did not allow to see that the units were in log-scale. The new figure is shown below. We hope this has clarified the sentence.

[Figure]

Line 118: Bottom right panel y-axis in Fig. S6 is cut off. What is being plotted on the x-axis – is this both phytoplankton functional group and chlorophyll? Can the x-axis label be adjusted to be more clear?

The new figure S6 (with no more cut-off) is given below. Overall this figure has the same set up as Fig. 4, but we do agree that the x-axes were misleading, since we showed here sums of the various groups and the / might have indicated ratios. We modified this and also to clarify this figure we used a similar figure caption as for Fig. 4:

[Figure]

*"Figure S6: Vertical distribution (0-50 m depth) of selected phytoplankton groups at sea-ice covered stations north of 80°N. According to Dybwad et al., (2021) stations 39, 43, 46 (Yermak Plateau) were in pre-bloom phase, while all other stations were in a bloom phase. Stations 19 and 32 were shelf stations. The contribution of the various phytoplankton groups is expressed as Chl a concentrations."*

Line 123: Should read "Supplement S7: "

changed

Lines 131-146: References should be alphabetized and formatting should be consistent.

Done

Line 133: Callahan et al. reference is missing a year

Done

---

## Author Comment (AC3)

Comments from the reviewer are in black; answer to the reviewer are in blue, new adding to the text are in black and italics.

Reviewer 3

The manuscript by Gros et al. reports on a large-scale survey of DMS, methanethiol and several other gases along the gradient between the eastern subpolar North Atlantic and the Arctic. Their main goal was to assess the concentrations and spatial distribution of these compounds and relate them to the environmental (physical, microbiological) conditions via statistical correlations. Based on their results, the authors aim to improve our current understanding on trace gas cycling in the context of a changing Arctic.

General assessment

The topic of this manuscript is certainly relevant for a wide biogeochemistry community and contributes to amend the large gaps of data coverage for marine trace gases; in particular for compounds which are understudied in comparison with other climate-relevant gases such as CO2 or CH4. The paper is very well written, its structure is clear and the methodological approaches are both sound and explained with enough detail. Although the manuscript is quite descriptive, the authors state clearly that they aim to report on the results of their survey. Hence, from that perspective, they were successful in achieving that. In my opinion a significant drawback of the study is the absence of air-sea fluxes of the different compounds measured in surface waters. Based on the content of the methods presented by the authors I cannot judge whether they have information at hand to do so, but it would be worth making an effort to provide estimates of air-sea fluxes for at least some of the compounds (I do think this can be done for DMS and CO).

We thank the reviewer for the overall positive evaluation of our manuscript and for the useful comments and suggestions. We agree that it would be very interesting to assess air-sea fluxes for the measured compounds and indeed our initial aims was also to perform measurements in the air, but it turned out that the parallel installed sampling line for atmospheric measurements aboard the vessel, dedicated for $CO_2$ measurements, had shown large contaminations. Thus, we decided to focus on ocean measurements only and in addition, we like to point out that we are here mainly focusing on the link between measured VOCs and biology (phytoplankton and bacteria). We think that adding some "theoretical" estimates of air-sea fluxes would go beyond the scope of this manuscript.

Other than that, most of my comments to the paper are minor (see below).

Specific comments

Title: The current title is rather long and can be misleading. The word "Variability" is should not be used generally here since the authors are addressing the spatial variability of several trace gasses, not their temporal variability.

The title is indeed quite long, but we do think it is important to mention the main elements. Following the suggestion of reviewer 2, we have added "dissolved" to the title, to avoid any confusion with atmospheric measurements. To address the suggestion of reviewer 3, we have now replaced "Variability" by "Concentrations":

*"Concentrations of dissolved dimethyl sulphide (DMS), methanethiol and other trace gases in context of microbial communities from the temperate Atlantic to the Arctic Ocean".*

29-30: Revise syntax, in particular after "understanding of". Also, I recommend stating how specifically the paper contributes to that understanding, since right now this could mean anything.

We changed the sentence as follows:

*Overall, the demonstrated latitudinal and vertical patterns contribute to understanding how concentrations of central marine trace gases are linked with chemical, and biological parameters across oceanic waters.*

80: Please replace the word "levels" for "concentrations" here and it all instances where you refer to that quantity.

Done

81: "numbers" is too unspecific. Please refer precisely to which measurable quantity you are referring to here.

We agree and have changed to *"concentrations"*

82: Bacterial diversity and water masses were addressed. No sea ice data cover was presented and therefore it should not be presented as a factor in your experimental design.

Sea ice cover was removed

88: The citation to Peeken (2016) is unnecessary. I know such reports have a doi number, but they do not constitute a source of peer-reviewed information and should only used when absolutely needed (see journal's regulations)

 The reviewer is correct that this reference is a report, but since it is the cruise report with valuable information about the expedition, we would prefer to keep it in the manuscript.

92: Stating that "usually" there is not sensor drift is not enough, even if the sentence is supported by a publication. The authors need to show that this was indeed the case during their survey in order to keep the credibility of their observations.

We do understand that the wording was a little misleading but please note, these tests are routinely carried out by the staff of Polarstern and we are only informed if the sensors are not behaving in the designated ranges. We thus wrote now:

*"The instrument performs a self-cleaning routine every day with acid washing and freshwater rinsing. In addition, sensor behavior is controlled by staff members of Polarstern (for details see Petersen (2014))".*

106-107: Here the citation is also unnecessary and should be removed. If the authors want to refer to the data used, there are better ways such a data set doi from Pangaea.

We disagree with the reviewer to delete this citation, since it credits the hard work people invest to create data products, which can be used by everybody if they need it for their studies. To credit the other colleagues involved in the CTD work we would like to keep this citation.

117: Delete point

Done

123: delete "l" after Chl a

Done

123: The R2 value in S1 is different than the one shown here. Revise.

The value given in S1 is R (0.91), the value given in the text is R2 (which have been changed from 0.84 to 0.83).

153-154: Revise wording. I would suggest "The measurement principle of PTRMS is (...)" or similar.

Done

214-217: The details on how this statistical analysis was setup should be explained in the "Material and Methods" section (i.e. independent and dependent variables, etc.). Otherwise the statement seems arbitrary (i.e. coming from nowhere).

We have added more information at the end of the bacterial method part:

*"Nonmetric multidimensional scaling was performed to determine bacterial community variability along the transect. Associations between the abundance of bacterial ASVs and environmental parameters were determined via Holm-corrected Spearman's correlations. Only correlations >|0.4| were considered, and only if higher than with latitude to omit indirect signals due to geographical variability"*

234-235: Explain the details on how the system was adjusted. This reads as if the authors used the continuous system for profiling. Was that the case? If so, a detailed description is needed.

During the transect, continuous measurements were performed; for the vertical profiling, measurements were made on samples collected using the CTD rosette. This has been described in the section 2.1 (see below)

"Along the ship track between May 19ᵗʰ and 27ᵗʰ, trace gases were continuously measured in the surface water layer. (…).After May 27ᵗʰ, eight ice stations (number 19, 27, 31, 32, 39, 43, 46, and 47, Table S1) were carried out (...) During the ice stations, discrete seawater samples for trace gas and phytoplankton composition analysis were collected at six different depths of the water column using the CTD (conductivity, temperature, depth) water-sampling carousel. These samples were collected in 1 L light-proof flasks for direct analysis on board."

As the reviewer refers to a sentence (line 234-235) which corresponds to the "results section", we do not think this experimental information should be given here. Nevertheless, for clarity we now refer to section 2.1.

259-266: I am not convinced of the approach here. Why was station 19 removed from the analysis? It appears that although stations 19 and 32 have high productivity, both isoprene and CO behave completely different. Also, if one compares CO concentration at stations 19 and 39 (having contrasting chl a concentrations), it becomes evident that CO is not affected by the same processes as other gases. The reasons for this are unfortunately not discussed at all. In order to explain the variability of some of the CO concentrations at depth (e.g. at stations 32 and 43), the authors claim that differences in the profiles are due to "decreased photochemical production following lower light penetration". However, this is the case for all stations and therefore it is not a compelling reason to explain the decrease with depth. Perhaps the authors rather refer to the effect of different sea ice coverage percentages in light penetration (?). If so, they can easily explore this possibility by using such data which is widely available.

We are very sorry, but we made a mistake when transferring the data from Excel to Sigmaplot for the final Fig 4 of the CO profiles, which was not measured at every station. Thus our results and discussion did not match completely the figure. We hope that our discussion about CO is now easier to follow. In addition we now clearly separate isoprene and CO in the discussion.

We also had the rationale of excluding station 19 for the correlation between isoprene and Chl a, since this is the only station where diatoms exclusively dominated the biomass. From our previous laboratory studies we did find that cold water diatoms emit less isoprene compared to temperate species (Bonsang et al. 2010) and thus we decided it was reasonable to exclude this station. This is now discussed as follows:

"Isoprene also markedly correlated with Chl a ($R^2$ = 0.6, Fig. S8), but only when excluding station 19. This correlation supports a biological source of isoprene, in line with the shown linkage of isoprene and Chl a maxima (Tran et al., 2013). Station 19 was the only station where diatoms dominated almost exclusively the phytoplankton biomass. As shown in laboratory

experiments, cold-water diatoms only emit little isoprene (Bonsang et al. 2010), which could explain the observed behavior.

In contrast to the latitudinal transect, MeSH showed low concentrations at most ice stations, except for station 19 (with higher concentrations and a clear decrease with depth). Station 19 was special since being located above the shelf and harbouring a diatom-dominated phytoplankton community it might be speculated that the diatom community also produces MeSH, but overall we have currently no real explanation as to why it is associated with a higher MeSH than the other stations.

The vertical profile of CO shows a decrease with depth as shown in Tran et al. (2013). This supports the notion that CO photoproduction (the main source of CO in the ocean) decreases up to threefold from the surface to 20 m depth (Fichot and Miller, 2010). An exception is station 31 where CO peaked at 30 m depth. This could indicate the presence of a large CO emitter, as the emission of CO can vary by more than an order of magnitude between phytoplankton species (Gros et al., 2009).

[Figure]

**Fig. 4.** Biological parameters and trace gas vertical distribution (0-50 m depth) at sea-ice covered stations north of 80°. According to Dybwad et al., (2021) stations 39, 43 and 46 (Yermak Plateau) were in a pre-bloom phase, while all other stations were in a bloom phase. Stations 19 and 32 were shelf stations. The contribution of each phytoplankton group is expressed as Chl a concentrations.

303: Is the mean value for surface waters or does it include the water column measurements? Please clarify.

It has been clarified as follows:

"For polar waters, the mean value of 5.9 ± 2.9 nM *in surface measurements during the transect* (...)"

308: Personal communications are not appropriate. Even less in this case since there are already two citations supporting the statement.

The personal communication has been removed

353: Same comment to personal communications. The authors already used Dybwad et al. (2021) as a defining criterion for the bloom stages in the study area at the time of sampling.

The personal communication has been removed. Instead we cite as suggested:

"Dybwad et al. (2021)"

385-387: This statement is contradictory with the results presented by the authors for CO. Based on the data presented it is only clear that CO production is not necessarily tied to a biological component and that photochemistry might have had a more significant role at the time of sampling. The authors argue (L.313-315) that low CO production by diatoms might be the explanation for the low concentrations at e.g. 19. However. this is speculative and cannot be substantiated with their observations. I recommend revising this aspect of the discussion.

We agree with the reviewer that the sentence "these probably have phytoplankton driven origins" does not apply to CO. Therefore, we have completed the sentence as :

Whereas isoprene, acetone, acetaldehyde and acetonitrile concentrations decreased northwards, CO, DMS and MeSH were uncorrelated with latitude and retained considerable concentrations in polar waters. Hence, these probably have phytoplankton-driven origins with regional variability, e.g. through localized blooms *and/or the presence of sea-ice.*

Supplementary information: there are inconsistencies in the naming of Figs. S6-S8. For instance, in L.235 S7 is mentioned although it does not match what is actually shown.

We are sorry about the confusion with numbering of figures S6 to S8. This has now been corrected.

In Fig. S8 no CO is shown (although announced in the main text) and the caption does not match the figure.

Indeed, this is a mistake in the main text, which should not mention CO when mentioning S8 (now S7). This has now been corrected. The figure caption of S8 (now S7) does correspond to the graphics (DMS and isoprene correlations with Chl a). As the figures S6 to S8 have now been correctly numbered, there should be no more confusion.

---

## Referee Report (RR1)

**Review of the paper: "Variability of dimethyl sulphide (DMS), methanethiol and other trace gases in context of microbial communities from the temperate Atlantic to the Arctic Ocean"**

by Valérie Gros et al.

**General comments**

The authors have undertaken an in-depth revision and have improved considerably most of the article. However, the writing could still be improved throughout, and I recommend a thorough check of English (grammar and orthography) and the References. The work looks solid from a technical standpoint, but I encourage the authors to report whether cross-calibration between underway and Niskin samples was performed. If not, please add a note of caution because Niksin sampling and underway systems can yield somewhat different concentrations, especially in the case of gases (at least DMS) that respond quickly to mechanical stress induced by underway pumping on cells.

A final note: this paper contains at least 2 different stories, and the one about DMS and MeSH has been treated more in depth and could make a separate paper. Of course this is the authors' decision.

**Specific comments**

L18: please indicate months-year.

L42: please support the "lifetime of 1 day" with a reference

L70: this new sentence is a bit repetitive, as the previous already said oceans can be a sink for OVOCs.

L90: maybe Simó et al. (2022), which is cited in the Discussion, is an appropriate citation here

L111: please check the excitation emission wavelengths, they look very weird. Chlorophyll does not absorb much at 325 nm. Rather, it has an absorption peak in the blue, at around 430 nm, and fluoresces in the red, at around 680 nm.

L122: please specify material.

L277, section 3.1.2: I suggest first mentioning the clear separation between bacterial communities from Atlantic and Polar water masses indicated by NMDS (Fig. 3).

L474: the studies of Kettle and Lawso reported mean DMS/MeSH ratios of around 5.5. This corresponds to a MeSH / (DMS + MeSH) of 15%, higher than the <10% indicated by the authors.

L486: cyanobacteria are well-known DMSP consumers, and are especially competitive under sunlight compared to heterotrophic bacteria. So It's not inconceivable that they produce MeSH as a DMSP assimilation by-product. Citations:

Malmstrom, R. R., et al. (2005). Dimethylsulfoniopropionate (DMSP) assimilation by Synechococcus in the Gulf of Mexico and northwest Atlantic Ocean. *Limnology and Oceanography*, *50*(6), 1924-1931.

Vila-Costa, et al. (2006). Dimethylsulfoniopropionate uptake by marine phytoplankton. *science*, *314*(5799), 652-654.

Ruiz-González, C., et al. (2012). Sunlight modulates the relative importance of heterotrophic bacteria and picophytoplankton in DMSP-sulphur uptake. *The ISME journal*, *6*(3), 650-659.

**Technical corrections and typos**

L80: Please correct: "This impact of sea-ice has been shown to DMSP"

L196: please check writing

L201: allows

L309: "for surface values" → "near the surface"

L333-341: check writing

---

## Referee Report (RR2)

Follow-up on Specific Comments

Line 201: How the MeSH sensitivity was determined presents uncertainty to the MeSH measurements, as the authors mention. Calculating the sensitivity as the average of sensitivities of compounds with similar m/z assumes that the sensitivity is primarily dependent on mass transmission. The authors state that the value used (13.4 ncps/ppb) is at the high range of measured calibration coefficients. It would be good to state what this range in calibration coefficients is. Other work that also has not had an experimentally measured calibration factor for MeSH has used the sensitivity to DMS given their similar collision rate constants and transmission efficiencies (Lawson et al. ACP (2020)). Another paper that measured calibration factors for MeSH and for DMS on a PTR-MS found that they were 3x more sensitive to DMS than MeSH (Novak et al. ACP (2022)). Since the MeSH findings in this paper are so important and have implications for the significance of MeSH in the sulfur budget, it would be good to have a more nuanced discussion of the uncertainty in MeSH based on what the range in calibration factor could be. I suggest this section of the supplemental provides a range in [MeSH] based on applying the average sensitivity of acetone and acetaldehyde, the DMS sensitivity, and the range in sensitivity coefficients in this study.

Line 202: Please provide information on how exactly this uncertainty was estimated. For example, what is the uncertainty in your calibrations, etc. and how does that lead to an overall value of +/- 20 or 30%. The provided reference Baudic et al. (2016) also does not provide this detail on how the uncertainty was calculated. This uncertainty estimation seems low given the lack of calibration to MeSH.

Line 363: I suggest moving the MeSH/(MeSH+DMS) values in Table 1 to a separate column. Also please clarify the unit for MeSH/(MeSH+DMS). This looks like percentage, if so, please add that to the column title.

---

## Author Response (AR2)

A point-by-point reply to the comments is given below.

Comments from the reviewer are in black; answer to the reviewer are in blue, new adding to the text are in black and italics.

**Review of the paper: "Variability of dimethyl sulphide (DMS), methanethiol and other trace gasesin context of microbial communities from the temperate Atlantic to the Arctic Ocean"**
by Valérie Gros et al.

**General comments**
The authors have undertaken an in-depth revision and have improved considerably most of the article. However, the writing could still be improved throughout, and I recommend a thorough check of English (grammar and orthography) and the References. The work looks solid from a technical standpoint, but I encourage the authors to report whether cross-calibration between underway and Niskin samples was performed. If not, please add a note of caution because Niksin sampling and underway systems can yield somewhat different concentrations, especially in the case of gases (at least DMS) that respond quickly to mechanical stress induced by underway pumping on cells. A final note: this paper contains at least 2 different stories, and the one about DMS and MeSH has been treated more in depth and could make a separate paper. Of course this is the authors' decision.

We thank the reviewer for the positive evaluation of our revised manuscript and hope the paper is now again written in a clear and well-structured way. In this new revised manuscript, we have performed a thorough check of English.

We agree that the paper mostly focus on DMS and MeSH with only little discussion on the other compounds. Nevertheless, we prefer to let the presentation of all compounds in this manuscript, as there are very few measurements in this high latitude zone.
There was no cross calibration between underway and Niskin samples. Therefore, following the reviewer's suggestion, we have added this note of caution in section 2.3:

"As no cross calibration was made between transect and Niskin measurements, possible differences between on-line and off-line measurements could not be evaluated. "

**Specific comments**
L18: please indicate months-year.
Done

L42: please support the "lifetime of 1 day" with a reference
The reference of Kloster et al. (2006) have been added

L70: this new sentence is a bit repetitive, as the previous already said oceans can be a sink for OVOCs.

We do not understand this comment, since this sentence shall highlight the fact that many OVOCs have a terrestrial source. We therefore like to keep this sentence.

L90: maybe Simó et al. (2022), which is cited in the Discussion, is an appropriate citation here
The citation has been added at the suggested place.

L111: please check the excitation emission wavelengths, they look very weird. Chlorophyll does not absorb much at 325 nm. Rather, it has an absorption peak in the blue, at around 430 nm, and fluoresces in the red, at around 680 nm.
We apologize for this. We now mention the correct values given by the company.
It reads now:

*"wavelengths of 460 nm and 620-715 nm respectively"*

L122: please specify material.
The material (glass) has been specified

L277, section 3.1.2: I suggest first mentioning the clear separation between bacterial communities from Atlantic and Polar water masses indicated by NMDS (Fig. 3).
We agree with the Reviewer's suggestion and have added the sentence
*"Accordingly, communities markedly varied between Atlantic and polar waters >80°N  (Fig. 3a)."*

L474: the studies of Kettle and Lawso reported mean DMS/MeSH ratios of around 5.5. This corresponds to a MeSH / (DMS + MeSH) of 15%, higher than the <10% indicated by the authors.

≤15% is now given in the text

L486: cyanobacteria are well-known DMSP consumers, and are especially competitive under sunlight compared to heterotrophic bacteria. So It's not inconceivable that they produce MeSH as a DMSP assimilation by-product. Citations:
Malmstrom, R. R., et al. (2005). Dimethylsulfoniopropionate (DMSP) assimilation by Synechococcus in the Gulf of Mexico and northwest Atlantic Ocean. *Limnology and Oceanography*, *50*(6), 1924-1931.
Vila-Costa, et al. (2006). Dimethylsulfoniopropionate uptake by marine phytoplankton. *science*, *314*(5799), 652-654.
Ruiz-González, C., et al. (2012). Sunlight modulates the relative importance of heterotrophic bacteria and picophytoplankton in DMSP-sulphur uptake. *The ISME journal*, *6*(3), 650-659.

We agree that cyanobacteria can take up DMSP, and have modified the text as follows including the citation to Vila-Costa et al.:
*"* The link between cyanobacteria and MeSH potentially relates to the known uptake of DMSP by *Synechococcus* and *Prochlorococcus* (Vila-Costa et al., 2006), although DMSP-utilizing genes are overall rare in cyanobacteria (Liu et al., 2018)."

**Technical corrections and typos**
L80: Please correct: "This impact of sea-ice has been shown to DMSP"

For clarification, it has been reworded to
"This influence of sea-ice has been shown for DMSP, DMS, isoprene, acetone and acetaldehyde in the Canadian Arctic (Galindo et al., 2014; Wohl et al., 2022; Galí et al., 2021)"

L196: please check writing

The repetition has been deleted, the sentence now reads:
Measurements were typically performed every 2.5 minutes, except between 61.1 °N to 65.3°N, where measurements were performed every 10 min for approximately 24 h, to scan a wider range of masses (m/z).

L201: allows
Done

L309: "for surface values" → "near the surface"
Done

L333-341: check writing

The paragraph now reads:

"DMS and Chl a were strongly correlated ($R^2$ Pearson's correlation coefficient = 0.93; Fig. S6). Isoprene also correlated with Chl a ($R^2$ = 0.6, Fig. S6) but only when excluding station 19. This correlation supports a biological source of isoprene, in agreement with previous demonstrated links between isoprene and Chl a maxima (Tran et al., 2013). Station 19 was the only station where diatoms almost exclusively dominated the phytoplankton biomass. The little emission of isoprene by cold-water diatoms (Bonsang et al., 2010) could explain this pattern.

In contrast to the latitudinal transect, MeSH showed low concentrations at most ice stations, with the exception of station 19. Station 19 was special due to its location above the shelf, and the phytoplankton community dominated by diatoms.

CO concentrations overall decreased with depth (Tran et al., 2013), except for  station 31 with a CO peak at 30 m depth. "

Reviewer 2

Follow-up on Specific Comments

Line 201: How the MeSH sensitivity was determined presents uncertainty to the MeSH measurements, as the authors mention. Calculating the sensitivity as the average of sensitivities of compounds with similar m/z assumes that the sensitivity is primarily dependent on mass transmission. The authors state that the value used (13.4 ncps/ppb) is at the high range of measured calibration coefficients. It would be good to state what this range in calibration coefficients is. Other work that also has not had an experimentally measured calibration factor for MeSH has used the sensitivity to DMS given their similar collision rate constants and transmission efficiencies (Lawson et al. ACP (2020)). Another paper that measured calibration factors for MeSH and for DMS on a PTR-MS found that they were 3x more sensitive to DMS than MeSH (Novak et al. ACP (2022)). Since the MeSH findings in this paper are so important and have implications for the significance of MeSH in the sulfur budget, it would be good to have a more nuanced discussion of the uncertainty in MeSH based on what the range in calibration factor could be. I suggest this section of the supplemental provides a range in [MeSH] based on applying the average sensitivity of acetone and acetaldehyde, the DMS sensitivity, and the range in sensitivity coefficients in this study.

To follow the reviewer's suggestion, we have now included the following paragraph in the supplement.

*"As MeSH quantification has been performed by using a sensitivity coefficient based on the average of the sensitivity of acetaldehyde and acetone, the uncertainty associated to this choice has been estimated. For this, the estimated coefficient has been compared on one hand to an averaged sensitivity coefficient and on the other hand to an estimated DMS sensitivity coefficient. For the first comparison, the average sensitivity coefficient of 9.4 ncps/ppb represents the mean of 10 sensitivity coefficient (determined for m/z 33, 42, 45, 59, 69, 71, 73, 79, 93 and 107). If we would apply such a sensitivity coefficient to MeSH, it would increase all concentration by a factor of 1,43. So for example, the mean value of MeSH for polar water would be 4,24 nM instead of 2, 96 nM. As mentioned, DMS was not in included in the standard that we had at that time. Recently, we have purchased a NPL (National Physics Laboratory, Teddington, UK) standard containing a series a compounds, including acetaldehyde, acetone and DMS. We have performed 3 calibrations (on different days) at the laboratory with the same PTRMS used during the TRANSSIZ campaign. By taking into account the ratio of the DMS sensitivity (14, 4 ncps/ppb) compared to an average of acetaldehyde-acetone sensitivity (21,2 ncps/ppb), we obtained a value of 1, 47, so very close from the first evaluation. Finally, we conclude that due to the absence of a calibrated standard for MeSH, concentrations reported in this paper could be underestimated by a factor of 1.5."*

Line 202: Please provide information on how exactly this uncertainty was estimated. For example, what is the uncertainty in your calibrations, etc. and how does that lead to an overall value of +/- 20 or 30%. The provided reference Baudic et al. (2016) also does not provide this detail on how the uncertainty was calculated. This uncertainty estimation seems low given the lack of calibration to MeSH.

Information is now provided in Supplement S4. It reads as follows

*"A complete estimation on the gas-phase measurement of this PTRMS has been performed in Baudic et al. (2016). This estimation, based on the ACTRIS measurement guidelines VOC 2014 (see Debevec et al., 2017), calculates the expanded uncertainty of U (X) as*

$$U(X) = k \times u(X) + DL_x/3$$

*With k being the coverage factor (here 2), u (X) the combined uncertainty in X, and $DL_x$ the detection limit of the species X. The combined uncertainty includes errors on standard gas, calibrations, blanks, reproducibility/repeatability, linearity, and relative humidity parameters. This expanded uncertainty has a maximum of 21% (21%, 18%, 9% and 10% for m/z 42, 45, 59 and 69 respectively). We do not give here the detailed contribution of each factor, as those calculations were not done specifically for this campaign. Nevertheless, we note that the two main sources contributing to the overall uncertainty were due to linearity error and to the uncertainty of VOC concentrations in the calibration standard gas. An additional uncertainty is the conversion of gaseous ppb into nM (based on the error of the calibration linearity, see calibration example for acetone in S4a). The overall uncertainty was then estimated at 21%, 32%, 11% and 11% for m/z 42, 45, 59 and 69 respectively. Therefore, the uncertainty for the calibrated compounds (including DMS, which was not present in the gas-phase standard but which has been calibrated with a liquid standard) has been estimated at about 30%.*

*As MeSH has been quantified using a sensitivity coefficient based on the average of the sensitivity of acetaldehyde and acetone, we assessed the uncertainty by comparing (i) to an averaged sensitivity coefficient, and (ii) to an estimated DMS sensitivity coefficient. For (i), the average sensitivity coefficient of 9.4 ncps/ppb represents the mean of 10 sensitivity coefficients (determined for m/z 33, 42, 45, 59, 69, 71, 73, 79, 93 and 107). If applying such a sensitivity coefficient to MeSH, it would increase concentrations by a factor of 1.43. As mentioned, DMS was not present in the standard that we had at that time. Recently, we have purchased a NPL (National Physics Laboratory, Teddington, UK) standard containing a series of compounds, including acetaldehyde, acetone and DMS. We have performed three calibrations (on different days) in the laboratory with the same PTRMS used during TRANSSIZ. Taking into account the ratio of the DMS sensitivity (14.4 ncps/ppb) compared to an average of acetaldehyde-acetone sensitivity (21.2 ncps/ppb) we obtained a value of 1.47, hence almost identical to the first evaluation. Overall, we conclude that due to the absence of a calibrated standard for MeSH, concentrations reported in this paper could be underestimated by a factor of ~1.5."*

We add here an information for the reviewer. New uncertainties estimations have been made for this PTRMS, in the framework of its long-term use at the ACTRIS-SIRTA station. These estimations, following updated guidelines from ACTRIS (which will be soon available), are described in detail in Simon et al. (2022), https://doi.org/10.5194/essd-2022-406. The overall uncertainty (taken into account precision and accuracy) has been estimated at 16%, 18%, 14% and 21% for m/z 42, 45, 59 and 69 respectively, consistent with the previous estimation.

The reviewer is right to point out a different uncertainty for MeSH. This has now been specified in the supplement (see above) and in the main text:

*"The overall uncertainty for dissolved VOC measurements was estimated at ± 30%, except for MeSH. Due to the missing direct calibration of MeSH, its concentration could be underestimated by up to 1.5 times (see S4). Therefore, reported concentrations presented here for MeSH have to be considered as lower limit."*

Line 363: I suggest moving the MeSH/(MeSH+DMS) values in Table 1 to a separate column. Also please clarify the unit for MeSH/(MeSH+DMS). This looks like percentage, if so, please add that to the column title.

A new column has been added to present MeSH/(MeSH+DMS) and the unit (%) has been added